# Patterns and drivers of dimethylsulfide concentration in the northeast Subarctic Pacific across multiple spatial and temporal scales

Alysia E. Herr[1], Ronald P. Kiene[2], John W. H. Dacey[3], Philippe D. Tortell[1,4]

[1]Department of Earth, Ocean and Atmospheric Sciences, University of British Columbia, Vancouver, BC, V6T 1Z4, Canada

[2]Department of Marine Sciences, University of South Alabama, Mobile, AL, 36688, USA

[3]Woods Hole Oceanographic Institute, Woods Hole, MA, 02543, USA

[4]Department of Botany, University of British Columbia, Vancouver, BC, V6T 1Z4, Canada

*Correspondence to*: Alysia E. Herr (aherr@eoas.ubc.ca)

**Abstract.**

The northeast subarctic Pacific (NESAP) is a globally important source of the climate-active gas dimethylsulfide (DMS), yet the processes driving DMS variability across this region are poorly understood. Here we examine the spatial distribution of DMS at various spatial scales in contrasting oceanographic regimes of the NESAP. We present new high spatial resolution measurements of DMS across hydrographic frontal zones along the British Columbia continental shelf, together with key environmental variables and biological rate measurements. We combine these new data with existing observations to produce a revised summertime DMS climatology for the NESAP, yielding a broader context for our sub-mesoscale process studies. Our results demonstrate sharp DMS concentration gradients across hydrographic frontal zones, and suggest the presence of two distinct DMS cycling regimes in the NESAP, corresponding to microphytoplankton-dominated waters along the continental shelf, and nanoplankton-dominated waters in the cross-shelf transitional zone. DMS concentrations across the continental shelf transition (range <1−10 nM, mean 3.9 nM) exhibited positive correlations to salinity ($r=0.80$), sea surface height anomaly (SSHA; $r=0.51$) and the relative abundance of prymnesiophyte and dinoflagellates ($r=0.89$). In contrast, DMS concentrations in near shore coastal transects (range <1−24 nM, mean 6.1 nM) showed a negative correlation with salinity ($r=-0.69$, $r=-0.78$) and SSHA ($r=-0.81$, $r=-0.75$), and a positive correlation to relative diatom abundance ($r=0.88$, $r=0.86$). These results highlight the importance of bloom-driven DMS production in continental shelf waters of this region, and the role of prymnesiophytes and dinoflagellates in DMS cycling further offshore. In all areas, the rate of DMS consumption appeared to be an important control on observed concentration gradients, with higher DMS consumption rate constants associated with lower DMS concentrations. We compiled a dataset of all available summertime DMS observations for the NESAP (including previously unpublished results) to examine the performance of several existing algorithms to predict regional DMS concentrations. None of these existing algorithms was able to accurately reproduce observed DMS distributions across the NESAP, although performance was improved by the use of regionally tuned-coefficients. Based on our compiled observations,

we derived an average summertime distribution map for DMS concentrations and sea–air fluxes across the NESAP, estimating
a mean regional flux of 0.30 Tg of DMS-derived sulfur to the atmosphere during the summer season.
**1 Introduction**
Spurred by a proposed role in climate regulation as a source of cloud-condensation nuclei and back-scattering aerosols, the
biogenic trace gas dimethylsulfide (DMS) and related organic sulfur compounds dimethylsulfonioproprionate (DMSP) and
dimethyl sulfoxide (DMSO) have been studied for more than four decades (Lovelock et al. 1972; Charlson et al. 1987).  This
body of research has revealed complex sulfur biogeochemical cycling in the oceans, and important physiological and
ecological roles for these molecules (Simó 2004; Stefels et al. 2007).  DMSP and DMS have been shown to play an essential
function in marine microbial systems as sources of carbon and sulfur (Kiene et al. 2000; Reisch et al. 2011).  These molecules
also act as olfactory foraging cues for numerous species of birds, fish, marine invertebrates and mammals (Seymour et al.
2010; Johnson et al. 2016), thereby driving interactions both within and beyond the marine microbial food web.  The ecological,
chemical and climatological significance of DMS and related compounds has stimulated significant effort to understand the
surface ocean distribution of these molecules and the underlying factors driving their variability.
The Pacific Marine Environmental Laboratory (PMEL) has compiled a database of over 47,000 discrete DMS measurements.
Lana et al. (2011, hereafter L11) utilized these data to construct a global climatology of surface ocean DMS concentrations
and sea–air fluxes, providing broad-scale understanding of oceanic distribution patterns.  The global mean DMS concentration
is estimated to be approximately 2 nM, but the climatology reveals several regional 'hot spots' of elevated DMS accumulation,
including polynya waters of the Southern Ocean, and the northeast Subarctic Pacific (NESAP).  In these regions, surface ocean
DMS concentrations 5–10-fold higher than the mean oceanic value are commonly observed (Kiene et al., 2007; Lana et al.
2011, Jarníková and Tortell 2016).  Although large-scale global patterns derived from the climatology are likely robust, a fuller
understanding of spatial and temporal patterns of regional DMS variability is constrained by the relatively poor spatial and
temporal coverage of existing measurements.
The NESAP, defined here as the region bounded by 44.5˚ N and 61˚ N latitude and 180˚ W and 120˚ W, exhibits consistently
high summertime DMS concentrations in both open ocean and coastal regions, with maxima of ~20 nM observed during the
late summer season (Wong et al. 2005; Asher et al. 2011, 2017; Steiner et al. 2012).  This oceanic region is also characterized
by strong spatial heterogeneity of environmental characteristics. High-productivity coastal upwelling regions transition to iron-
limited high nutrient low chlorophyll (HNLC) waters offshore (Boyd and Harrison 1999; Boyd et al. 2004).  Seasonally varying
surface currents, fresh water inputs, coastal upwelling and recurrent formation of westward-propagating mesoscale eddies
result in semi-permanent and transient hydrographic frontal zones, impacting regional marine biodiversity and productivity

(Crawford et al. 2005; Whitney et al. 2005; Ribalet et al. 2010). This spatial heterogeneity makes it challenging to quantify DMS distributions from discrete ship-based sampling, and complicates region-wide generalizations of DMS dynamics.

Recent work has highlighted differences in the distribution of DMS and related compounds across distinct domains of the NESAP, particularly in offshore and coastal regions (Wong et al. 2005; Asher et al. 2011, 2017; Steiner et al. 2012). The HNLC offshore region was identified by L11 as an area of high DMS concentrations and sea−air fluxes. Results from in situ observations (Wong et al. 2005; Levasseur et al. 2006; Merzouk et al. 2006; Asher et al. 2011) and numerical models (Steiner et al. 2012) suggest that elevated DMS concentrations in these open ocean waters are driven by the presence of high DMS/P producing phytoplankton taxa, such as prymnesiophytes and dinoflagellates, and the effects of mixed layer stratification and Fe-limitation, which may act to increase DMS/P production as a means to offset oxidative stress (Sunda et al. 2002, Kinsey et al. 2016). A low particulate organic carbon to sulfur ratio in the HNLC regime further influences bacterial DMSP metabolism, resulting in increased DMS-yield from DMSP metabolism (Merzouk et al. 2006; Royer et al. 2010). In the physically dynamic coastal waters of the NESAP, high DMS concentrations likely result, in part, from seasonal coastal upwelling, which drives high phytoplankton biomass accumulation. Recent work (Asher et al. 2017) has demonstrated an enhancement of DMS accumulation following upwelling events in the coastal NESAP, consistent with previously observed high DMS/P concentrations in other upwelling regions (Hatton et al. 1998; Zindler et al. 2012; Wu et al. 2017). Increased DMS concentrations in the post-upwelling bloom phase may result from nitrogen limitation, increased grazing pressure (which releases DMSP into the dissolved pool; Simó et al. 2018), oxidative stress associated with shoaling mixed layers, and a phytoplankton community shift towards high DMSP-producing species (Nemcek et al. 2008; Franklin et al. 2009). Despite these advances in understanding DMS dynamics in the NESAP, many aspects of DMS cycling in this region remain poorly documented, including the factors influencing interannual variability (Steiner et al., 2012; Galí et al., 2018), the interplay between iron concentration and phytoplankton community shifts (Levasseur et al. 2006; Roayer et al. 2010), and the relative importance of phytoplanktonic DMSP lyases and micrograzers (Steiner et al. 2012).

New advances in sensor technology over the past decade have begun to significantly expand DMS data coverage in a number of ocean regions. These fine scale measurements reveal novel features and highlight the apparent influence of oceanographic frontal zones in driving fine-scale DMS distribution patterns (Holligan et al. 1987; Locarnini et al. 1998; Belviso et al. 2003; Tortell 2005a; Nemcek et al. 2008; Royer et al. 2015; Jarníková et al. 2018). In previous work (Asher et al. 2017), we have documented sharp transitions in DMS concentrations across salinity frontal zones in nearshore NESAP waters. This earlier work did not include corresponding measurements of DMS/P turnover rates, limiting mechanistic interpretation of the observed spatial patterns. To our knowledge, there has been no systematic evaluation of the processes driving fine-scale DMS variability across frontal zones. Such a study requires high resolution concentration measurements together with assessments of biological productivity and DMS/P turnover rates.

In this article, we present a new data set of DMS/P concentrations across coastal and open ocean waters of the Subarctic
Pacific, from the northern Gulf of Alaska to the Oregon coast. Using a suite of measurements collected during two summer
cruises (2016–2017), we document regional-scale features, and characterize sub-mesoscale DMS structure across
hydrographic frontal zones in on-shelf and transition regions. Using real-time ship-board measurements, we were able to
select contrasting sites across frontal zones for more extensive sampling and analysis, allowing us to probe underlying rate
processes in adjacent areas with distinct DMS/P concentrations and surface water hydrography. We combined our new data
set with existing observations from our own group and from the existing PMEL database to produce a new summertime DMS
climatology for the NESAP. This updated climatology enables us to better constrain the summertime distribution of DMS in
the NESAP, identifying persistent 'hot spots', and exploring correlations between DMS concentration and other biotic and
abiotic variables. We also use our compiled data set to evaluate various empirical algorithms predicting DMS concentrations
and sea–air fluxes across the NESAP. Our results yield new insights into the spatial patterns and potential drivers of
summertime NESAP DMS distribution across various spatial scales in a globally important oceanic region.
**2 Methods**
**2.1 Data overview**
In this study, we combined new data from two recent oceanographic expeditions with existing observations derived from
several decades of compiled DMS measurements in the NESAP. Ancillary measurements of various environmental and
biological variables were obtained from a number of sources (ship-based measurements, remote sensing and blended data
products) to help interpret DMS distribution patterns. The various data sets are described below.
**2.2 New high-resolution data sets**
**2.2.1 Underway ship-board measurements**
Field sampling was conducted on board the University–National Oceanographic Laboratory System (UNOLS) vessel *Oceanus*
during July of 2016 and August of 2017 (O16, O17, respectively). Our cruise tracks included offshore, coastal and transitional
waters throughout the Gulf of Alaska (Fig. 1). We define the coastal regime as those waters with bottom depths shallower
than 2000 m, following Asher et al. (2011). We utilized real-time DMS measurements (see below) and NASA satellite ocean
colour imagery (AquaMODIS) to guide our cruise track, enabling us to identify areas with high concentrations of DMS and
strong spatial gradients in surface water phytoplankton biomass and hydrography (sea surface temperature and salinity).
During O16 we also conducted detailed surveys of three hydrographic frontal zones that exhibited sharp DMS concentration
gradients. One of these surveys (T1; Fig. 1) was located in the coastal-open ocean transition near Dixon Entrance north of
Haida Gwaii (formerly the Queen Charlotte Islands), while the other two transects were located along the British Columbia
continental shelf (T2: Hecate Strait and T3: La Perouse Bank; Fig. 1). After an initial survey to examine frontal structure,
stations were selected for depth-resolved sampling to cover the gradients present across the frontal zone. The O17 cruise
covered a similar area as O16. Although we did not perform detailed transect surveys on this second cruise, we did sample
waters near T1−T3.
High resolution surface water DMS measurements were conducted using membrane inlet mass spectrometry (MIMS)
following published methods (Tortell 2005b; Nemcek et al. 2008). The MIMS system, sampling from the ship's underway
seawater flow through system (~5 m intake depth), allows for high-frequency measurements (2−3 times per minute), yielding
a spatial resolution of ~150–200 m at normal ship speeds of 8−10 kts. During these cruises, DMS concentrations were also
measured in discrete water samples collected at 5 m depth using a purge-and-trap system connected to a gas chromatograph
equipped with a flame-photometric detector (FPD-GC) (Kiene and Service 1991). These discrete measurements were used to
assess the accuracy of MIMS-based measurements. We found good agreement between methods, with a mean absolute error
of 0.90 nM, root mean square error of 1.4 nM, and coefficient of determination of $r^2=0.89$ between the two instruments across
the full range of measured concentrations (Fig. 3).
High resolution DMS measurements were paired with rate measurements and ancillary underway data to examine potential
drivers of spatial variation. A ship-board thermosalinograph was used to measure sea surface temperature (SST) and salinity
at high spatial resolution (SBE 45 and SBE 38 for salinity and temperature, respectively). Chlorophyll-a (chl-a) concentration
was measured using a WET labs ACS absorbance/attenuation meter, based on the absorption line height at 676 nm (Bricaud
et al. 1995; Roesler and Barnard 2013; Burt et al. 2018). These chl-a concentrations were further used to derive an estimate
of phytoplankton assemblage size structure and taxonomic distributions, based on the empirical algorithm of Hirata et al.
(2011). Phytoplankton size-class estimates derived from this algorithm agreed well ($r^2 >0.75$) with discrete HPLC-derived
estimates (methods described below; Zeng et al. 2018). MIMS was also used to determine the ratio of oxygen and argon
concentrations relative to atmospheric saturation. The resulting biological oxygen saturation term, $\Delta O_2/Ar$, can be used to
calculate net community productivity (NCP) from the air−sea gas exchange of $O_2$ (Kaiser et al. 2005). We used the calculation
approach of Reuer et al. (2007) to compute NCP from our $\Delta O_2/Ar$ measurements. We note that some of these estimates,
particularly in regions of active upwelling, are likely negatively biased by the entrainment of $O_2$ under-saturated water into the
mixed layer. While this effect can be accounted for using $N_2O$ measurements (Izett et al. 2018), we do not have these data
available for our cruises. Our derived NCP estimates thus likely represent under-estimates, and we have removed all negative
NCP values. Nonetheless, the general spatial patterns we observed in NCP are likely to be robust.
**2.2.2 Station-based measurements**
We measured a suite of variables at selected sampling stations along the cruise track. All water for ancillary measurements
was taken from 5 m depth, collected using Niskin bottles. A Seabird CTD probe (Seabird 911plus) was deployed at each
station to obtain depth profiles of hydrographic features over the upper 200 m of the water column. A density difference
criterion of 0.05 kg m$^{-3}$ was used to calculate mixed layer depths.
DMS loss and DMSP consumption rates were measured using the radio-labeled $^{35}$S methods outlined by Kiene and Linn (2000)
with some modifications to minimize the release of DMSPd during incubations. Briefly, $^{35}$S-labeled DMSPd or DMS were
added to samples at non-perturbing concentrations (<1 % of ambient levels). Samples were incubated in the dark at surface
water temperatures for <1 h ($^{35}$S-DMSP) or <7 h ($^{35}$S-DMS). The rate constant for DMSPd turnover was determined by
measuring the disappearance of $^{35}$S-DMSP from the dissolved (< 0.2 µm) pool. The rate constants for DMS loss were
determined by measuring the accumulation of dissolved, non-volatile $^{35}$S transformation products derived from the volatile
$^{35}$S-DMS tracer. Consumption rates (nmol L$^{-1}$ d$^{-1}$) were calculated by multiplying in situ DMS or DMSPd concentrations by
the measured rate constant (k$_{DMS}$ or k$_{DMSPd}$ respectively).
Primary productivity was measured using 24 h $^{14}$C uptake incubations, following the method outlined by Schuback et al.
(2015). Incubation bottles were held in a deck-board incubator plumbed with continuously flowing seawater to achieve in situ
temperature. The light intensity was adjusted to ~ 30 % surface irradiance enriched in blue light using neutral density screening
in combination with blue photographic film (LEE filters: #209 and CT blue maximum transmission at approximately 460 nm).
Light levels in the tank were measured with a ULM-500 light meter equipped with a 4π-sensor (Walz). Bacterial production
was measured using the tritiated leucine method (Smith and Azam 1992) and converted to carbon units (Simon and Azam
1989; Ducklow et al. 2000). Station samples were also analysed for total and dissolved DMSP (DMSPt and DMSPd) with a
GC-FPD discrete method using the previously described NaOH cleavage and small-volume gravity drip filtration method
(Dacey and Blough 1987; Kiene and Slezak 2006). DMSPp was calculated by subtracting DMSPd from DMSPt (Zindler et
al. 2012, Levine et al. 2016).
We obtained discrete estimates of phytoplankton assemblage composition using diagnostic pigment analysis (DPA) of
photosynthetic pigments measured using HPLC. For these measurements, 1 L samples were collected on GF/F filters (nominal
pore size ~ 0.7 µm), flash frozen in liquid nitrogen and stored frozen until analysis at the NASA Goddard Space Flight Center
Ocean Ecology Laboratory (Van Heukelem and Thomas 2001). The DPA method was originally developed by Vidussi et al.
(2001), and subsequently refined (Uitz et al. 2006; Hirata et al. 2008; Brewin et al. 2010) to more accurately capture
phytoplankton type and size class. The estimation formulas used here are those of Hirata et al. (2011), with coefficients tuned
specifically for the NESAP by Zeng et al. (2018). Percent contribution to phytoplankton assemblage was assessed for three
size classes (micro, nano, and pico).

**2.3 Compilation of published data**

To provide a broader regional spatial context for our observations, we combined discrete DMS measurements from the PMEL data archive with high spatial resolution DMS measurements made using MIMS since the early 2000s. Table 1 provides dates and spatial domains of the cruises, along with relevant literature citations. Note that some of the DMS data included in this compilation have not been previously published. All of our compiled MIMS data have been made available on the PMEL database (https://saga.pmel.noaa.gov/dms/).

**2.3.1 MIMS data sets**

MIMS-based observations included in this study are derived from 11 cruises conducted between 2004 and 2017, primarily aboard the Canadian Coast Guard Ship *John P. Tully* as part of ongoing time-series monitoring programs conducted by the Department of Fisheries and Oceans Canada (DFO). Only summertime data (defined here as June, July and August; JJA) falling within the NESAP region (44.5˚–61˚ N, 180˚–120˚ W) were included in this compilation. Although DMS concentrations and phytoplankton biomass often remains high through September (Galí et al., 2018; Lana et al., 2011; Steiner et al., 2012), there are fewer DMS data available for this month. Measurements were binned to a temporal sampling resolution of 1 minute. All DMS data points are paired with shipboard sea surface salinity and SST. The cruises VIJ04, VIJ10, WCAC10, LPA11, O16 and O17 also include paired NCP estimates obtained from MIMS measurements, using the $\Delta O_2$/Ar-based method described above.

**2.3.2 PMEL data extraction**

We accessed the PMEL data base (http://saga.pmel.noaa.gov/dms/) on 6 December, 2017 to extract observations from June, July and August in the NESAP region defined above. Our selection criteria yielded 3236 data points between 1984 and 2003. These observations were relatively evenly distributed between the three months, but were biased spatially, with a preponderance of data derived from on-shelf waters off the coast of Alaska (see Fig. 8b). As with MIMS data, the majority of data points in the PMEL data base included paired sea surface salinity and SST measurements (94.6% and 99.8%, respectively).

**2.4 Ancillary measurements**

Ancillary oceanographic data were used to contextualize DMS spatial distributions, examine potential correlations to environmental variables and evaluate the performance of several empirical algorithms predicting DMS concentrations. In many cases, ancillary variables of interest (e.g. chl-*a*) were not reported in conjunction with DMS data, and we thus utilized a number of remote sensing data products, as described below. Remotely-sensed parameters were linearly interpolated to the spatial resolution of ship-based DMS observations.

AquaMODIS satellite data were used to obtain information on photosynthetically available radiation (PAR; Frouin et al. 2003),
chl-*a* (OCI algorithm; O'Reilly et al. 1998; Hu et al. 2012), calcite (Gordon et al. 2001; Balch et al. 2005) and diffuse
attenuation coefficients (Werdell and Bailey 2005). For these data products, we extracted level 3 gridded data from
http://oceancolor.gsfc.nasa.gov/cgi/l3 at 9 km resolution. Monthly means for chl-*a*, calcite and $k_d$ were utilized to maximize
spatial coverage by minimizing data gaps caused by cloudiness, whereas 8 day average PAR data were used. AquaMODIS
chlorophyll and sea surface temperature (SST) data were also used to estimate sea surface nitrate (SSN) using a North Pacific-
specific algorithm (Goes et al. 2000). Aqua MODIS data are only available starting in July of 2002, whereas most of the
PMEL data set in this region is from sampling prior to 2003. For earlier observations (going back to 1997), we used chl-*a* data
from the SeaWiFS satellite. Satellite chl-*a*, calcite and $k_d$ data were unavailable for data prior to 1997 (<1% of DMS data).
We obtained information on sea-surface height anomalies (SSHA) using gridded data sets (5 day, 0.17° x 0.17° resolution)
obtained from ftp://podaac-ftp.jpl.nasa.gov/allData/merged_alt/L4/cdr_grid_interim. This level 4 satellite product is derived
from various sensors, and data are not available before 1992. Mixed layer depths at a monthly, 1° resolution were obtained
from the China Second Institute of Oceanography (CSIO) ftp://data.argo.org.cn/pub/ARGO/BOA_Argo/. These data are based
on gridded Argo float data interpolated using the Barnes method, and are available for the years 2004–present (Li et al. 2017).
Due to limitations in Argo operational depths, data are largely absent from waters shallower than 2000 m (136 out of 249 1° x
1° bins).
We calculated sea–air DMS fluxes from DMS concentration data and surface wind-speeds using the gas transfer
parameterization of Sweeney et al. (2007) and the Schmidt number formulation of Saltzman et al. (1993). Wind speed data
for       flux       calculations       were       obtained       from       the       NCEP/NCAR       reanalysis       dataset
(https://www.esrl.noaa.gov/psd/data/gridded/data.ncep.reanalysis.pressure.html) at a 2.5° daily resolution. These calculations
were performed prior to data binning (described below), such that temporally-resolved sea–air flux was calculated for all
~150,000 DMS data points. Following previous studies, we assume negligible atmospheric DMS concentrations for our
calculations, leading to a potential (though likely small) overestimate of the sea–air flux. For purposes of comparison to fluxes,
we calculated DMS column burden along transects by multiplying DMS concentration and average mixed layer depth.
**2.5 Data binning and province assignment**
High resolution, underway measurements may introduce sampling biases due the large number of data points collected. For
example, a ship holding station will increase spatial data density at a particular location, and the large number of observations
can exert a disproportionate influence on derived mean values. To address this, all measurements in the data set were assigned
to 1° spatial bins, in which all observations for a given year were averaged. All observations within the JJA months for a given
year were averaged, rather than deriving separate monthly climatologies. The resulting yearly data grids were then averaged
to create long-term gridded means. This technique effectively assigns equal weight to each year of measurements in a given
grid cell. Both DMS and paired ancillary parameters were binned using this method.
Following the approach of L11, data grid cells were assigned to Longhurst Biogeochemical Provinces to examine patterns
across different regimes within the greater NESAP (Longhurst 2007). Three primary provinces fall within the domain of our
study region: California Upwelling Coastal Province (CCAL), Alaska Downwelling Coastal Province (ALSK), and Pacific
Subarctic Gyres Province – East (PSAE) (Fig. 8). The CCAL province as defined by Longhurst extends south to 16.5° N.
Hereafter, all references to the CCAL refer to the portion of this province above 44.5° N latitude. Province boundary
designations were obtained from www.marineregions.com (accessed October 2017), and the MATLAB native inpolygon.m
function was used to assign grid cells to individual provinces. Any grid cell either inside or on the edge of boundaries was
assigned to a particular province. As such, some data cells (37 out of 249 total) are assigned to multiple provinces. Average
summer DMS concentrations and flux measurements were computed for each province. For comparison to L11, we
recalculated the average summertime DMS concentration and flux in the three study provinces using only the PMEL data
utilized by L11. The PMEL data were first binned using the year-weighted method described above.
**2.6 Statistical analysis and empirical algorithms**
We used our compiled data set to examine broad-scale relationships between DMS and other oceanographic variables. For
this analysis, data were log-transformed to overcome non-normal distributions, and the strength of pair-wise relationships was
assessed by computing Pearson's correlation coefficients. Correlations were applied to 1˚ x 1˚ binned data both within
individual provinces and across the entire NESAP.
We also used several existing empirical algorithms to reconstruct DMS fields at a 1˚ x 1˚ resolution from various environmental
predictor variables, comparing the accuracy of the resultant products against our binned DMS observations. The algorithms
tested in this study include those of Simó and Dachs (2002), Vallina and Simó (2007), Watanabe et al. (2007), and Galí et al.
(2018) (hereafter, SD02, VS07, W07, and G18, respectively). Both SD02 and VS07 used global data bases to develop their
algorithms. SD02 relates DMS to chl-$a$:MLD, with chl-$a$ values > 15 µg L$^{-1}$ removed prior to analysis. VS07 relates DMS
concentration to solar radiative dose (SRD). This term, as defined by the authors, is based on light extinction coefficients ($k_d$),
sea surface irradiance ($I_0$), and mixed layer depth. Due to the large areal extent of the study area, we used AquaMODIS derived
PAR in lieu of the station-based $I_0$ measurements used by the authors. Similarly, strong variation in $k_d$ in coastal vs. open
ocean waters is expected. We thus modified the author's approach and used satellite derived $k_d$ (based on a chlorophyll-
dependent algorithm; Werdell and Bailey 2005) rather than a fixed coefficient. W07 uses data specific to the North Pacific
and relates DMS to SST, SSN and latitude. The two-step G18 algorithm utilizes a previously developed DMSPt predictive
algorithm based on chl-$a$ and MLD (Galí et al. 2015), in conjunction with PAR measurements. In order to test this algorithm,
we utilized the satellite-derived PAR, MLD and chl-$a$, described above. We further modify the author's approach by testing
performance on our 1° x 1° binned data, rather than data binned at a 5° x 5°, in order to maximize the number of observations.
Recognizing the utility of re-parameterizing proposed algorithms for specific areas, we tested algorithms using both published
linear coefficients, and coefficients derived specifically for the NESAP using a least-squares approach to determine best fit to
our data set. The coefficients used to test the original G18 were those regionally tuned by the authors for latitudes above 45°
N.

## 3 Results

We begin by presenting an overview of our new DMS measurements and ancillary data from the 2016–2017 summer cruises,
highlighting DMS distributions and the presence of distinct surface water properties across different parts of our transect. We
then provide a detailed description of DMS dynamics across several hydrographic frontal zones, discussing the potential role
of various processes in driving these gradients. Finally, we present an updated summertime climatology for this region,
compiling our new measurements with existing DMS observations from across the NESAP to examine large-scale patterns in
DMS distributions, and correlations with other oceanographic variables. The potential role of these variables in driving DMS
distributions in the NESAP, and the need for additional process studies is addressed in the discussion.

### 3.1 Oceanographic conditions in the NESAP during summer 2016–2017

Our 2016 and 2017 cruises surveyed oceanographic regimes from offshore HNLC regions to productive coastal upwelling
zones. As indicated by AquaMODIS satellite imagery, chl-*a* concentrations exhibited strong gradients across the oceanic-
coastal transition in both 2016 and 2017 (Fig. 1). Coastal waters showed elevated chl-*a*, with maximum values of 50 µg L$^{-1}$
and 18 µg L$^{-1}$ in 2016 and 2017, respectively. In both years, highest chl-*a* values were observed in waters with shallow mixed-
layer depths (<10 m) along the La Perouse Bank (Fig. 1). In the off-shelf regions, chl-*a* concentrations appeared uniformly
low in 2016, although significant cloud cover limited the availability of satellite imagery. By comparison, we observed
generally higher chl-*a* concentrations in offshore waters in 2017. Most notably, our cruise track passed through an apparent
coccolithophore bloom in the northern Gulf of Alaska, where a large calcite signal (~2 mmol PIC m$^{-3}$) was detected in
AquaMODIS imagery. Patterns in NCP were generally similar to those of chl-*a*, with elevated production in coastal waters
(Fig. 2c). In both years, we observed NCP on La Perouse Bank exceeding 100 mmol $O_2$ m$^{-2}$ d$^{-1}$ (Fig. 2c, inset).
Coastal regions exhibited generally fresher surface waters and shallower mixed layer depths, except for several regions of
enhanced vertical mixing associated with upwelling. This coastal upwelling signature was apparent in elevated salinity and
decreased temperature of surface waters, and also through the presence of negative sea surface height anomalies (Fig. 1c,d).
Small-scale regional heterogeneity in coastal regions was apparent in both years, with salinity and temperature exhibiting sharp
gradients over the continental shelf, associated with riverine input and complex mixing processes. By comparison, oceanic
surface waters showed less spatial heterogeneity, and were generally more saline, with deeper mixed layers (Fig. 2b). The
sea-surface height anomaly field indicated the presence of several Sitka and Haida eddies in both years (Fig. 1c,d), enhancing
mesoscale variability through the transport of coastal water offshore.
Using the approach of Hirata et al. (2011) and Zeng et al. (2018), we derived high resolution estimates of phytoplankton
assemblage composition from our underway chl-*a* measurements. This approach revealed a predominance of phytoplankton
in the micro- size class (>20 µm) in coastal waters (Fig. 2d), with an average of 50 % of chl-*a* attributable to
microphytoplankton. In contrast, off-shelf waters showed greater diversity in phytoplankton composition. In these waters,
microphytoplankton accounted for ~25 % of total chl-*a*, while the pico- and nano- size classes accounted for ~30 % and ~40
%, respectively (Fig. 2e,f).
**3.2 DMS distributions**
Across our study region, surface water DMS concentrations ranged from <1−24 nM in 2016 and <1−18 nM in 2017 (Fig. 2a,
Fig. 3). We observed a number of localized DMS 'hot spots' in regions of elevated chl-*a* and NCP. In both years, these
localized high DMS regions were particularly evident in the vicinity of the highly productive La Perouse Bank (Fig. 2a, inset).
We also observed several areas where strong DMS gradients co-occurred with salinity fronts. These areas include the T1−T3
transects survey in O16, detailed below. Despite associations between DMS concentration and several variables in some
localized regions, we only observed weak correlations between DMS and other measured variables across the full cruise tracks.
During O16, DMS concentrations were most strongly correlated to NCP, with a Pearson's coefficient of r=0.42 (p<0.001).
This relationship was substantially weaker in O17 (r=0.29, p<0.001).
**3.3 Detailed surveys of DMS across hydrographic frontal zones**
During the O16 cruise, we sampled along three repeated transects to map DMS distributions near hydrographic frontal zones.
All three transects showed significant gradients in salinity, chl-*a* and DMS/P concentrations, as well as in the metabolic activity
of phytoplankton and bacteria (Fig. 4, 6−7). While DMS concentrations appeared to co-vary with salinity and chl-*a* across
these frontal zones, the strength and direction of these relationships were not consistent across the three transects. We discuss
each transect in detail below.
**3.3.1 Transect 1**
T1 was located west of Dixon Entrance (Fig. 1) in waters influenced by the Alaska Current and coastal water masses. Offshore
waters along this transect were more saline and colder than those on the shelf. The area exhibited DMS concentrations up to
10 nM in off-shelf, saline waters (Fig. 4). Particulate and dissolved DMSP ranged from ~60–125 nM and ~1.8–4.7 nM,
respectively, and showed no significant correlation to DMS (Fig. 4d). At the shelf break (approx. 134.4˚ W, indicated on Fig.
4 by dotted line), we measured a sharp drop in salinity and corresponding decrease in DMS concentrations, with concentrations
remaining below ~3 nM over the most coastal parts of the transect. Across the entire T1 transect, DMS concentrations
displayed a striking fine-scale coherence to salinity (r=0.80, p<0.001; Fig. 4a,b). A significant positive correlation was also
observed with SSHA (r=0.51, p<0.001), indicating a potential influence of westward-propagating Haida eddies. Fig. 5 shows
a line plot of SSHA measurements from the approximate time of T1 sampling, overlaid by DMS concentrations. The coherence
between DMS concentrations and mesoscale oceanographic features can be seen in this figure despite differences in the spatial
resolution of the two data sets.
The lower salinity coastal waters along T1 were characterized by elevated chl-*a* concentrations (Fig. 4c), resulting in a negative
correlation between DMS concentrations and chlorophyll (r=-0.47, p<0.001). Figure 4c shows the estimated percent
abundance of diatoms and combined dinoflagellates and prymnesiophytes as derived from HPLC-based DPA-analysis. The
remaining phytoplankton assemblage consisted largely of picoeukaryotes (13 − 36 %). Although HPLC samples are not
available for all of the coastal waters we sampled, results obtained from the empirical algorithm of Hirata et al. (2011) using
underway absorption data suggest a shift in phytoplankton assemblage composition from smaller size classes in offshore waters
to a microphytoplankton-dominated community in on-shelf waters. DMS exhibited relatively weak, though statistically
significant (p<0.001) positive correlations with the algorithm-derived relative abundance of nano- and picophytoplankton size-
classes (r=0.55 and r=0.38, respectively), and a negative correlation with the relative abundance of microphytoplankton (r=-
0.53). In support of this result, discrete HPLC measurements revealed a strong positive relationship between DMS
concentration and the combined relative abundance of prymnesiophytes and dinoflagellates (r=0.89, p=0.001), and a negative
correlation to diatom abundance (r=-0.70, p=0.036). We also observed a strong positive correlation between DMS and
DMSPp:chl-*a* (r=0.80, p=0.003) suggesting higher cellular DMSP concentrations in phytoplankton assemblages in the off-
shelf regions of this transect. Overall, results from this transect demonstrate a transition from high DMS concentrations in the
lower productivity, nanophytoplankton dominated offshore waters, to low DMS concentrations in higher productivity, diatom-
dominated nearshore region.
Average rate constants (d$^{-1}$) for biological consumption of DMS and DMSPd appeared qualitatively higher in the on-shelf
region (although insufficient sampling does not allow for reliable statistical testing), suggesting faster removal of DMS/P from
coastal surface waters with lower DMS concentrations. For DMS and DMSPd respectively, loss constants averaged 1.15 ±
0.3 d$^{-1}$ and 88.2 ± 13.9 d$^{-1}$ onshore, as compared to 0.66 ± 0.045 d$^{-1}$ and 39.6 ± 1.45 d$^{-1}$ in offshore stations (Fig. 4e). Net
primary productivity and bacterial productivity also showed a qualitative trend towards higher average values in the low DMS
coastal waters, but these differences were not statistically significant. Although biological loss of DMS constitutes only one
of several loss terms, the patterns observed here suggest that enhanced microbial activity and relatively higher DMS/P
consumption rate constants played a role in maintaining lower concentrations of these compounds in nearshore waters.
We calculated the mixed layer DMS burden by multiplying concentration and average mixed layer depth (13 m). Biological
DMS loss integrated over the mixed layer averaged 22 $\mu$mol m$^{-2}$ d$^{-1}$, sufficient for daily removal of 47 % of the DMS burden.
By comparison, derived sea–air flux estimates across the transect exhibited a mean value of 13 $\mu$mol DMS m$^{-2}$ d$^{-1}$, accounting
for ~ 25 % of the mixed layer DMS burden daily. Due to a relatively homogenous wind field over the area of our sampling
transect, the sea–air fluxes were tightly correlated to DMS concentrations, such that the lower DMS concentrations in nearshore
regions cannot be explained by greater rates of ventilation to the atmosphere.
**3.3.2 Transect 2**
The second sampling transect, T2, was located in the coastal waters of Hecate Strait situated on the continental shelf (Fig. 1).
Sea surface temperatures along this transect exhibited low variability (standard deviation ~0.5˚ C), with the coldest waters
located mid-transect in areas of highest chl-*a*. Mixed layer depths ranged from 10–15 m, and DMS concentrations ranged
from < 0.5 nM to nearly 20 nM (Fig. 6). DMSPd concentrations exhibited only minor variations over the transect (2.5–2.8
nM), while DMSPp concentrations showed greater variability (61–144 nM). Neither
DMSPd or DMSPp were correlated to DMS concentrations. HPLC measurements suggested that diatoms dominated across
the entire transect, and particularly in northern regions (Fig. 6c). Picoeukaryotes and green algae comprised the bulk of the
remaining phytoplankton assemblage composition (generally < 10 %). In contrast to our observations for T1, DMS
concentrations exhibited negative correlations to both salinity (r=-0.69, p<0.001; Fig. 6b) and SSHA (r=-0.81, p<0.001) in this
area and were not significantly correlated to chl-*a* (Fig. 6c). Despite the lack of correlation to chl-*a,* DMS did exhibit
significant, though weak, positive correlations with estimates of relative microphytoplankton abundance (r=0.22, p<0.001),
and stronger negative correlations with the abundance of pico- and nano- size classes (T2: r=-0.47, r=-0.45; p<0.001; Fig. 6c).
In support of this observation, HPLC-pigment data from discrete sampling stations revealed a strong positive relationship
between DMS concentration and relative abundance of diatoms (r=0.88, p=0.001), and a negative correlation between DMS
and combined dinoflagellate and prymnesiophyte abundance (r=-0.88, p=0.001). These correlations suggest diatoms as an
important source of DMS, in contrast to that observed for T1.
Unlike bulk chl-*a* concentrations, we found that primary productivity showed a strong positive correlation with DMS along
T2 (r=0.90, p=0.037), although this result is based on only four data points. Bacterial productivity was also significantly higher
in the high DMS waters, although this variable was even more sparsely sampled along the transect, and we cannot infer any
meaningful statistical association with DMS (Fig. 6f). As with T1, both k$_{DMSPd}$ and k$_{DMS}$ appeared higher in the low-DMS
portions of the transect. Across the entire transect, DMS and DMSP consumption rate constants ranged from 0.51 to 1.29 d$^{-1}$
and 28.8 to 49.5 d$^{-1}$, respectively (Fig. 6e). This result suggests microbial consumption as potential driver of DMS
distributions, with higher DMS/P consumption rate constants in waters with lower DMS concentrations.
Integrated biological DMS loss was significantly higher than that of T1, with an average 78 µmol m$^{-2}$ d$^{-1}$ (equivalent to removal
of 87 % of the DMS burden per day). By comparison, DMS sea-air flux across the transect was low, with a mean value of 2.9
µmol m$^{-2}$ d$^{-1}$. This flux was sufficient to remove only ~6 % of mixed layer DMS burden daily. We thus conclude that biological
processes play a significant role in DMS turn-over along this transect.
**3.3.3 Transect 3**
T3 was located in the highly productive coastal waters of La Perouse Bank, along the continental shelf of the west coast of
Vancouver Island (Fig. 1). DMS concentrations across this transect ranged from <1–24 nM, while DMSPd and DMSPp
concentrations were among the highest observed cruise-wide (1.1 – 9.8 nM and 26 – 480 nM, respectively; Fig. 7d). DMSPp
was correlated to DMS (r=0.76, p=0.02). Mixed layer depths ranged from 8–12 m, with the shallowest values found in fresher,
salinity-stratified inshore waters influenced by riverine input. Sea surface temperature was lower in these low salinity waters,
although it varied little over the transect (standard deviation < 1° C). With respect to other measured variables, DMS behaved
similarly to the coastal T2 transect (Fig. 7a). We observed negative correlations between DMS and salinity (r=-0.78, p<0.001;
Fig. 7b) and SSHA (r=-0.75, p<0.001). We also found elevated chl-*a* in the low salinity waters, although there was only a
weak positive correlation between chl-*a* and DMS (r=0.25, p<0.001) across the full transect (Fig. 7c).
Microphytoplankton consisting primarily of diatoms dominated the low-salinity, high-DMS waters of the transect, with a shift
towards smaller cells observed in the more saline waters farther offshore (Fig. 7c). Among the T3 stations, green algae,
prokaryotes, and picoeukaryotes each comprised ~5 – 20 % of phytoplankton abundance. Similar to T2, we found a significant
positive correlation between DMS and microphytoplankton (r=0.90, p<0.001), and a negative correlation between DMS and
phytoplankton of the nano- and pico- size class (r=-0.77, r=-0.75; p<0.001). In support of this observation, HPLC-pigment
data showed a strong positive relationship between DMS concentration and relative abundance of diatoms (r=0.94, p<0.001),
and a negative correlation with combined dinoflagellate and prymnesiophyte abundance (r=-0.74, p=0.023). A negative
relationship was also observed between DMSPp:chl-*a* and DMS (r=-0.88, p=0.002) (Fig. 7d). In contrast to T1, high DMS
coincided with regions of lower cellular DMSP concentrations among phytoplankton, consistent with the dominance of
diatoms in the high DMS portions of this transect.
Along the T3 transect, DMS exhibited a positive qualitative association with primary productivity and bacterial productivity,
though these relationships are based on very few sampling points. It is noteworthy that the bacterial productivity measured
along T3 was higher than anywhere else along the cruise track, with production more than 5-fold greater than the cruise-wide
average. Values of $k_{DMS}$ ranged from 0.8–2.7 d$^{-1}$ across the transect. As with T1 and T2, $k_{DMS}$ was higher in low-DMS regions
of T3. In contrast, $k_{DMSPd}$ values along T3 increased in parallel with DMS concentrations (higher rate constants in higher DMS
waters). DMSP loss constants ranged from 38.6 to 92.1 d$^{-1}$ (Fig. 7e). The highest DMSP loss constant translates into a derived
turnover time of just 16 minutes, and coincided with the highest bacterial productivity (~26 µg POC L$^{-1}$ d$^{-1}$).
Biological DMS loss integrated over the mixed layer was sufficiently high to remove >100 % of the DMS burden daily (~47
µmol m$^{-2}$ d$^{-1}$).  By comparison, sea–air fluxes were a minor loss term by comparison (4.9 µmol m$^{-2}$ d$^{-1}$), and were sufficient to
remove only ~12 % of the mixed layer DMS burden.  Due to low removal rates and relative homogeneity of wind speed fields,
sea–air flux cannot be invoked to explain the spatial distribution of DMS across this transect.
**3.4 Regional DMS distribution – comparisons of 2016 and 2017 observations with past studies**
To explore potential regional-scale relationships between DMS concentrations and other environmental variables, we
combined our new DMS data with measurements collected over the past three decades, including previously unpublished high-
resolution MIMS data.   The addition of new measurements to the existing PMEL data set substantially increases spatial and
temporal coverage in the NESAP.  When data were binned to 1˚ x 1˚ resolution, coverage was increased by ~20 % in the
CCAL and ALSK Longhurst provinces, and 14 % in the PSAE, with the overall addition of 90 data-containing grid cells (Table
2).  As shown in Fig. 8, our measurements primarily increase data coverage in waters below 57˚ N.  These regions were
previously under-sampled in the PMEL data set utilized by L11, which was strongly biased to measurements near the coast of
Alaska.  Figure 9a further illustrates the latitudinal shift in data coverage with the inclusion of additional MIMS data.  As
shown in Fig. 9b, average derived DMS concentrations across latitudinal bands at the north and south extremes of our study
area remain similar to those derived from the PMEL data set utilized by L11.  However, in the region between 50˚ N and 54˚
N, where there were few observations in the PMEL database, our compiled data show mean concentrations as much as 4.5 nM
(~40 %) lower than those calculated using PMEL data alone.
Table 3 shows the change in province-wide average DMS concentration, sea–air fluxes, and total summertime DMS flux based
on our updated analysis.  Relative to our revised estimates, DMS concentrations and sea-air fluxes derived using only the
PMEL data were lower in the CCAL and higher in both the PSAE and ALSK provinces.  The most pronounced difference was
that of sea–air flux in the PSAE, where estimated values decreased by 4.5 µmol m$^{-2}$ d$^{-1}$ (20 %).  Despite these regional
differences, the total summer DMS flux across the NESAP differed by only 6.5 % between our compiled data set (0.30 Tg S)
and the PMEL data set utilized by L11 (0.32 Tg S).
Our compiled data set provides greater confidence in DMS concentrations and sea–air fluxes across the NESAP, and enables
us to better constrain spatial patterns.  Figure 10 shows binned average summertime DMS concentration across the region, as
well as the derived sea–air DMS fluxes.  The highest concentrations were observed in ALSK, where coastal waters contain
maximum DMS concentrations exceeding 20 nM.  A persistent region of elevated DMS concentrations was also evident in
mid-PSAE oceanic region, with concentrations greater than 10 nM.  Sea–air DMS fluxes showed a spatial distribution similar
to DMS concentrations, with maximum values of >100 $\mu$mol m$^{-2}$ d$^{-1}$. Calculated DMS:chl-*a* ratios for binned data (Fig. 10c),
showed generally higher values in offshore NESAP waters.

**3.5 Correlations and algorithm testing**

Using our new data compilation, we examined the relationship between DMS concentrations and a suite of oceanographic
variables across the NESAP. Table 4 shows both NESAP-wide and province-specific correlations derived from this analysis.
While many correlations are weak or not statistically significant, some patterns do emerge, particularly in the offshore waters
of the PSAE domain. No single variable explains a large portion of the DMS variation in this province, but statistically
significant correlations exist between DMS and chl-*a* and calcite (r=0.45 and r=0.50, respectively). We also found a negative
relationship between DMS and SSHA (r=-0.47). For the ALSK province, we found weak inverse correlations between DMS
and SST (r=-0.32) and water depth (r=-0.34). Significant positive correlations between DMS and derived surface NO$_3$
concentrations, PAR, and chl-*a* are also observed (r=0.30, r=0.41, and r=0.34 respectively). In contrast to other provinces, we
observed a statistically significant correlation between DMS and NCP in the CCAL province (r=0.43). The lack of other
significant correlations in the province may, in part, reflect the lower number of data points obtained for this region.
Moving beyond simple pairwise correlations, multi-variate empirical algorithms provide an additional approach to assess the
potential drivers of regional DMS dynamics. We evaluated the ability of five previously published algorithms to reproduce
patterns in the DMS observations. In order to obtain the best possible results, we modified the original equations using a least
squares method to obtain the best-fit coefficients for our data set. We evaluated the algorithm outputs against observations
using Pearson's correlation coefficients and root mean square errors (RMSE). As shown in Table 5, model performance was
generally low, with most correlation coefficients less than 0.53 and RMSE values ranging from 1.2 to 81.6 nM. The best
results were obtained for the CCAL province, where both the tuned SD02 and the original G18 algorithms were able to predict
DMS concentrations with moderate success (r = 0.62*, RMSE = 1.61 and r = 0.72*, RMSE = 1.9, for SD02 and G18,
respectively). As both of these algorithms rely on MLD, which is only available for waters deeper than 2000 m, it is important
to note that predictive strength can only be assessed for these off-shelf waters, and should not be taken to represent performance
in coastal waters. The customized VS07 (with coefficients tuned to the NESAP data) showed the best overall performance
across the entire NESAP region. Yet, even this model showed only weak correlation between predicted and observed DMS
values (r=0.31). Notably, the original linear coefficients for this model yielded DMS concentrations that were inversely
correlated to the measured values. In no case did models using original linear coefficients outperform those using recalculated
coefficients.

## 4 Discussion

The results presented here provide new information on the fine-scale and regional patterns of DMS distributions across the NESAP. Our ship-board observations document sub-mesoscale variability in DMS concentration across hydrographic frontal zones, with associated process measurements providing insight into potential driving factors. By combining these new data with more than three decades of DMS measurements, we are able to improve data coverage for the NESAP to examine larger-scale spatial patterns and provide a more robust regional climatology to evaluate empirical predictive algorithms.

### 4.1 Contrasting cycling regimes within the NESAP

A number of studies have documented differences in DMS dynamics across oceanographic regimes in the NESAP (Royer et al. 2010, Asher et al. 2011, 2017). These regional differences result from complex ecosystem and environmental interactions, and limit broad-scale prediction of DMS concentrations and sea−air fluxes (Galí et al. 2018). Taxonomic composition of phytoplankton assemblages has been identified as a main driver of DMS distribution patterns. For example, dinoflagellates and prymnesiophytes typically have elevated DMS production, associated with greater intracellular concentrations of DMSP (Keller 1989) and, in some cases, high activity of DMSP lyase (the enzyme that cleaves DMSP to DMS and acrylate; Steinke et al. 2002; Wolfe et al. 2002; Curson et al. 2018). In contrast, bloom-forming diatom species have typically lower intracellular DMSP levels (Keller 1989), with the exception of some polar species (Levasseur et al. 1994, Matrai and Vernet 1997). However, nutrient limitation has been shown to significantly increase diatom DMS/P production (Bucciarelli and Sunda 2003; Sunda et al. 2007; Harada et al. 2009). Thus, the accumulation of DMS in the water column depends on both the composition of phytoplankton assemblages and their physiological state, as previously shown by Gabric et al. (1999). Other factors, including zooplankton grazing and the metabolic demands of heterotrophic bacteria are also important (e.g. Levasseur et al. 1996, Kiene and Linn 2000, Merzouk et al. 2006, Asher et al. 2017). Below, we discuss the potential factors driving high DMS concentrations along three frontal zones exhibiting sharp DMS concentration gradients. Specifically, we contrast the nanoplankton dominated T1 transect with the diatom-dominated coastal T2 and T3 transects, examining the environmental and biological conditions that may have led to the different DMS accumulation in these areas.

### 4.2 The importance of phytoplankton assemblage composition

The T1 transect, located in the southern-most portion of the ALSK province, spanned 5˚ of longitude from deep (>3000 m) offshore waters, into nearshore waters over the continental shelf. These oceanographic regimes were separated by strong hydrographic frontal features in the vicinity of the shelf break. The negative correlation between DMS and chl-*a* along this transect demonstrates that DMS accumulation did not directly scale with bulk phytoplankton biomass. Rather, our results suggest that DMS concentrations were likely influenced by phytoplankton assemblage composition, with the highest DMS concentrations associated with the greatest relative proportion of prymnesiophytes and dinoflagellates (Fig. 4c) and the highest DMSPp:chl-*a* (Fig. 4d). Similar relationships have been documented in numerous studies focusing on offshore waters of the

NESAP and elsewhere (e.g. Barnard et al. 1984; Hatton et al. 1999; Royer et al. 2010; Steiner et al. 2012). In these areas,
elevated DMS concentrations are often attributed to a preponderance of high-DMSP phytoplankton taxa.
Comparison of T2 and T3 with T1 shows that the association of elevated DMS with prymnesiophyte and dinoflagellate
dominance and high DMSPp:chl-*a* ratios did not hold across our entire survey region. As was observed by Royer et al. (2010),
we measured generally low DMSPp:chl-*a* ratios in the diatom-dominated coastal waters of T2 and T3 (Fig. 6d, 7d). Yet, DMS
concentrations measured in these waters were extremely high, at times exceeding 20 nM (Fig. 2a). Unlike the T1 transect,
DMS concentrations along T2 and T3 increased with decreasing DMSPp:chl-*a* ratios, and were strongly correlated with diatom
abundance.
One potential explanation for the difference between T1 and T2/T3 may relate to the different location of these sampling
regions. The T1 transect sits along the transition between offshore and inshore waters, where different nutrient regimes control
phytoplankton productivity. Inshore waters over the continental shelf are typically limited by macronutrients, whereas offshore
waters transition into iron-limitation (Boyd and Harrison 1999). At the boundary between these regimes, mixing of water
masses through horizontal advection can stimulate phytoplankton productivity (Lam and Letters 2008). Ribalet et al. (2010)
observed an active community of nanoplankton in the transitional waters, and attributed this to the stimulation of (often high-
DMSP) oceanic phytoplankton by water mass mixing, at the boundary of macro- and micro-nutrient rich waters.
Formation of Haida and Sitka eddies may aid in this mixing through the westward transportation of micronutrient-replete
coastal water (Johnson et al. 2005; Whitney et al. 2005). SSHA measurements can be used as an indicator of eddy-induced
mixing in this region, as warm-core Haida and Sitka eddies waters manifest as closed circulation features exhibiting positive
SSHA (Fig. 1c, d). We observed the highest DMS associated with positive SSHA along T1 (Fig. 5), suggesting the influence
of water mass mixing in driving mesoscale patterns of DMS distribution. Beyond this mesoscale coherence, unresolved sub-
mesoscale variability is likely attributable to biological heterogeneity (Fig. 4c, d; Royer 2015 et al.)
In contrast to the transition waters, nearshore waters over the continental shelf are typically dominated by low DMSP-
producing diatoms. Elevated DMS in these diatom-rich waters may reflect a combination of high absolute biomass and an
upregulation of DMSP production observed under nutrient stress (Bucciarelli and Sunda 2003; Sunda et al. 2007; Hockin et
al. 2012; Bucciarelli et al. 2013). A meta-analysis by McParland and Levine (2018) reported an average 12-fold upregulation
of intracellular DMSP production under nutrient-stress conditions among phytoplankton, including diatoms, typically
considered low-producers. By comparison, high DMSP producers only showed an average 1.4-fold upregulation. Our results
are similar to those of Barnard et al. (1984), who observed a decreasing influence of prymnesiophyte abundance on DMS
concentrations in the Bering Sea with increasing proximity to the continental shelf. We note that increases in
microphytoplankton abundances are often accompanied by increases in other phytoplankton groups (including high-DMSP
producing taxa; Barber and Hiscock 2006; Uitz et al. 2006). However, we observed no correlation between DMS and the
absolute abundance of prymnesiophytes and dinoflagellates for either T2 or T3 transects. This result suggests that while these
high-DMSP producing taxa may play a role in driving DMS concentrations along these transect, diatoms are likely dominant
contributors, as judged by the strong correlation between DMS and absolute diatom absolute abundance (r=0.91, p=0.001)
along T3.
In coastal waters, seasonal upwelling may drive high phytoplankton biomass accumulation and increased DMS production in
the late-bloom phase, when stratified surface layers are exposed to higher mean light intensities (due to shallow mixing) and
become nutrient depleted (Zindler et al. 2012). These environmental conditions would act to increase cellular oxidative stress,
thus promoting the production of DMS/P as part of a cellular response mechanism (Sunda et al. 2002). Further, several studies
have shown increased bacterial activity and higher rates of cellular DMSP leakage in the late-bloom phase (Malin et al. 1993;
Stefels and Boekel 1993; Matrai and Keller 1994). The results of Asher et al. (2017) demonstrating high DMS concentrations
in post-upwelling waters of the coastal NESAP support this idea. Measurements of SSHA in coastal regions can provide a
signature for recent upwelling; the combined effect of wind-induced seasonal water transport offshore and the presence of high
density (cold and saline) upwelled water acts to depress sea surface height relative to annual means (Smith 1974; Tabata et al.
1986; Strub and James 1995; Saraceno et al. 2008; Venegas et al. 2008). Negative relationships between DMS concentrations
and SSHA were observed in both the T2 and T3 transects, suggesting an association between DMS and upwelling events.
Additional ecosystem processes may influence DMS accumulation in surface waters. In particular, zooplankton grazing and
viral infection may increase DMS concentrations, due to the release of cellular DMSP in phytoplankton during sloppy feeding
and cellular lysis (Dacey and Wakeham 1986; Belviso et al. 1990; Hill et al. 1998). Both of these factors are density-dependent,
and thus likely to become more significant with higher phytoplankton cell densities in the late bloom phase. Unfortunately,
we do not have measurements to address these processes directly, but the elevated DMSPd concentrations along T3 (~7 nM)
may reflect viral and zooplankton mediated loss of particulate DMSP into the dissolved pool.
Taken together, our results support previous studies showing the importance of DMSP-rich species in driving high DMS
concentration in offshore waters of the NESAP and elsewhere (e.g Stefels et al. 2007, Royer et al. 2010, Steiner et al. 2012,
Asher et al. 2017). In coastal waters, it appears that diatom-dominated phytoplankton assemblages can also support elevated
DMS accumulation, particularly under high biomass conditions during the late bloom phase, as has been previously observed
in the Southern Ocean (Turner et al. 1995) and the Barents Sea (Matrai and Vernet 1997).
**4.3 The effect of DMS/P consumption rate on DMS distribution**
DMS consumption rates constants across our study area can be translated to biological DMS turnover times ranging from 9 h
to 2.5 d (average of 25 h). By comparison, turnover times calculated from sea−air flux removal rates averaged 6.1 d across
this area, suggesting that this term is less important in the mixed layer DMS budget. We note that DMS concentration is set

by the dynamic balance between production and loss terms (Galí and Simó, 2015), of which only a subset were measured in our study. Gross DMS production, DMS production from DMSP cleavage, and DMS loss from photo-oxidation, which we did not measure, constitute potentially important terms in driving DMS distribution. Further, our conclusions are limited by data coverage, and based, at times, on few measurements. Notwithstanding these limitations, to our knowledge, no study has yet assessed DMS/P turnover rates across frontal zones on the small spatial scales examined here. Our limited measurements thus remain important in comparing meso- and submesoscale processes to those operating on larger spatial scales. While these measures do not encompass all loss processes, we found that biological consumption and sea–air flux alone were sufficient to quickly erase a DMS accumulation signature in the mixed layer. Thus, DMS concentrations measured here appear to be reflective of short-term production and consumption processes.

Across our study area, biological DMS removal rate constants ($d^{-1}$) were inversely related to DMS concentrations (r=-0.55, p=0.03), with lower $k_{DMS}$ in waters with elevated DMS. This study-wide trend supports the relationships observed along each transect. The relationship may reflect a time-lag of bacterial response to increased DMS concentrations. Results from previous studies in other regions have shown that bacterial DMS consumption increases after a rapid rise in DMS concentrations, resulting in consumption rate constants that are relatively low when DMS concentrations are initially high. As consumption rate constants increase, DMS concentrations decrease (Zubkov et al. 2004; del Valle et al. 2009). These results, along with the observed positive correlation between DMS and bacterial activity (r=0.53, p=0.03), suggest that microbial consumption is an important control on DMS accumulation, irrespective of phytoplankton community assemblage. However, the positive correlation between DMS loss rates and concentrations suggests that microbial consumption may not be sufficient to offset new DMS production. Previous studies in other regions have examined the impact of DMS loss and production in driving distributions, demonstrating correlations between DMS concentrations and microbial consumption and production rates in some systems (Wolfe and Kiene 1993; Zubkov et al. 2002 Merzouk et al. 2006, Vila-Costa et al. 2008). The relationship observed here between DMS, $k_{DMS}$ and bacterial activity may reflect the preponderance of on-shelf stations measured for DMS consumption in our survey (10 out of 16 stations), and significantly higher rates of bacterial metabolism in onshore waters ($7.81 \pm 3.0$ vs $1.10 \pm 0.3$ µg POC $L^{-1}$ $d^{-1}$ for on- and off-shelf stations, respectively).

Recent studies in the NESAP have estimated that photo-oxidation may account for 20–70 % of gross DMS removal in the NESAP (Asher et al. 2017), and it is possible that this process is particularly important in offshore waters. Bouillon and Miller (2004) found that quantum yields of DMS photo-oxidation in the NESAP correlated well to nitrate concentrations, suggesting that this pathway is particularly relevant in the HNLC region where excess macronutrients persist throughout the summer. Thus, the role of biological DMS consumption on influencing total DMS concentrations may be more important in the generally low nitrate coastal waters.

Rates of DMSPd turnover were among the highest measured anywhere, likely due, in part, to the very high productivity of the
waters we sampled. However, no correlation was found between DMSPd loss rates or loss rate constants and DMS
concentrations in our study. This lack of correlation may be due, in part, to variation in DMSPd loss pathways. The DMS
yield of DMSP metabolism can vary significantly depending on metabolic needs of bacteria present, and relative abundance
of phytoplankton with DMSP lyase activity (Yoch 2002). Although DMS yield was not measured in this study, previous
reports have shown that in the NESAP, a low carbon to organic sulfur ratio in the HNLC regime results in increased DMS-
yield from DMSP metabolism, whereas onshore DMS-yield is relatively lower (Merzouk et al. 2006; Royer et al. 2010).
Further, variation in DMS loss processes may obscure a relationship between DMSPd cleavage and DMS concentrations, as
high loss terms may disproportionately impact net DMS production. We are currently investigating, in greater detail, the
patterns of DMS and DMSPd consumption from our O16 and O17 cruises (Kiene et al., in prep).
**4.4 Insights from merged data set**
Our merged data set, binned to 1° x 1° spatial resolution, builds on the L11 climatology to further constrain summertime DMS
distributions across the NESAP region. Despite an overall ~20 % increase in data-containing bins, and the inclusion of data
from seven additional years, we see only small changes in the derived climatological DMS concentrations and sea−air fluxes
when compared to the PMEL data set used by L11 (Table 3). Our new observations thus support the validity of the L11
climatology in the NESAP region, providing further confidence in the apparent distribution patterns, and a greater spatial
footprint for the climatological field. A significant result of our analysis is the presence of high DMS:chl-*a* in offshore waters
(Fig. 10c). This result builds on previous reports of higher DMSP:chl-a concentrations in offshore NESAP waters, and
highlights the importance of prymnesiophytes, dinoflagellates, and other DMSP-rich phytoplankton taxa in driving DMS
accumulation in this region.
**4.5 Biogeochemical provinces**
When examining results from our 1° binned data set, a separation of the NESAP into on- and off-shelf regimes does not capture
the biogeochemical complexities of the region. Ecological provinces, as defined by Longhurst (2007), define regions with
coherent seasonal trends in physical processes, which give rise to similar biological and chemical characteristics. The use of
Longhurst's biogeochemical provinces may thus be a more suitable (though still imperfect) approach to examine large-scale
and long-term differences in DMS cycling across the region. Work by Reygondeau et al. (2013) has demonstrated the potential
for shifts in province boundaries over time, including decrease of coastal province size during El Nino periods, and a general
shore-ward shift of ALSK boundaries during summer months. A model-based classification of marine ecosystems in the North
Pacific by Gregr and Bodtker (2007) divides our study region into six domains that show little similarity to Longhurst
provinces. It is difficult to say which of these classification schemes is most appropriate for examination of DMS dynamics.
However, for the sake of direct comparison with L11, we chose to use Longhurst's provinces to examine regional cycling
differences (Hind et al. 2011; Belviso et al. 2011; Royer et al. 2015). While we acknowledge these provinces provide only a
crude distinction of biogeochemical regimes, they remain a best-approximation without delving into more complicated time-
resolved ecological province models (Reygondeau et al. 2013). Going forward, it may be useful to examine DMS dynamics
in sub-regions defined with a number of different metrics.

**4.6 Correlation with environmental variables**

Our analysis shows that no single variable can explain an appreciable amount of variability in DMS concentrations across the
NESAP. This result is consistent with previous global and regional studies (Kettle et al. 1999; Vézina 2004; Lana et al. 2011).
Nonetheless, an examination of the differing relationships between DMS concentrations and other environmental variables
provides insight into potential underlying factors driving DMS distribution (Table 4). For example, although we found a
moderately strong significant positive correlation between DMS and chl-$a$ in the largely HNLC PSAE province, no relationship
was observed between these variables in the CCAL province. As noted above and confirmed in several previous studies, the
phytoplankton community structure in the offshore PSAE region consists largely of small, DMSP-rich species (Booth et al.
1993; Suzuki et al. 2002; Royer et al. 2010; Steiner et al. 2012), and large blooms are infrequent. Indeed, the average binned
chl-$a$ concentration in this province is $< 1$ µg L$^{-1}$. As such, modest increases chl-$a$ likely reflects a stimulation of this high
DMSP-producing community. The positive correlation with calcite (an indicator of high-DMSP producing coccolithophores)
supports this idea.
The relationship between chl-$a$ and DMS is more complicated in the CCAL. High productivity in coastal upwelling zones
results in a strong onshore/offshore trend in average chl-$a$ concentrations. Yet, no such trend is observed in DMS
concentrations. This may be due, in part, to the sensitivity of DMS concentrations to phytoplankton assemblage composition
and bloom dynamics. High phytoplankton biomass alone will not result in elevated DMS in this region. Rather, elevated
DMS concentrations may occur as a response to conditions of late-bloom nutrient stress, as discussed above and in section 4.7.
Factors driving observed DMS distribution patterns in the ALSK province are more difficult to surmise. DMS is notably high
in the cold, productive waters adjacent to the Alaskan Peninsula. This is affirmed by a weak negative correlation between
DMS and SST, and the positive correlation between DMS and chl-$a$. Given that this portion of the province is known to
experience localized summer upwelling, it is possible that high DMS in the regions simply reflects elevated productivity and
related upwelling-induced stressors.

**4.7 Algorithm performance**

Our results suggest that no single empirical algorithm is likely to perform well in predicting DMS distributions across the
subarctic Pacific, although some predictive success was observed in the offshore waters of the CCAL province. Perhaps the
most informative result was the negative correlation between measured and modelled results using the VS07 algorithm. This
algorithm predicts DMS concentrations from solar radiative dose, a term that measures depth-integrated exposure to sunlight.

The underlying assumption in this algorithm is that increases in SRD are accompanied by increases in DMS due to UV-induced oxidative stress (Vallina and Simó 2007). However, it is also possible that elevated SRD can also lead to a decrease in surface water DMS concentrations through DMS photo-oxidation. As observed in previous studies, photo-oxidation in the NESAP may account for up to 70 % of gross DMS removal, and rates are positively correlated with nitrate concentrations (Bouillon and Miller 2004; Asher et al. 2017). Thus, in the high-nitrate NESAP, SRD may serve primarily to remove DMS from surface waters, rather than stimulate DMS production, as suggested by the negative correlation between DMS and SRD across the NESAP and within the PSAE province (Table 5). In areas with low surface water nitrate concentrations, such as the CCAL province (Boyd and Harrison 1999), SRD could act to promote DMS accumulation. The good performance of the G18 algorithm in the CCAL province supports this idea. In contrast to the VS07 algorithm, G18 includes terms representing both irradiance (PAR) and biology (DMSPt estimate), thus including the influence the combined effect of biomass and phytoplankton physiological state. The poor performance of the G18 algorithm across other NESAP regions may be due to the nitrate–photolysis relationship described above, or to the limited environmental-dependence of DMSP production in prymnesiophyte / dinoflagellate-dominated HNLC phytoplankton (McParland and Levine 2018).

The results discussed above underline the need for regional algorithm tuning, and the selection of models best suited for a given area and season. There is a particular need to develop approaches representing DMS distributions in HNLC regions. In order to accomplish this goal, it will be important to improve mechanistic understanding of DMS/P dynamics, merging field-based process studies with prognostic numerical models (e.g. Aumont et al. 2002, Clainche et al. 2004, Steiner et al. 2012, Wang et al. 2015, Hayashida et al. 2016).

**5 Conclusion**

This study examines the distribution and cycling of DMS across the NESAP at various spatial scales. Our results confirm the importance of high-DMSP producers (i.e. prymnesiophytes and dinoflagellates) to DMS accumulation in offshore waters, while also demonstrating the importance of diatom-dominated assemblages in driving DMS distribution in coastal upwelling regions. We further highlight the importance of metabolic rate processes in DMS distributions, providing evidence for the importance of DMS consumption on concentration gradients at a fine-scale. On the short spatial scales covered by our transect surveys, we observed strong correlations between DMS concentrations and other variables (i.e. SSHA, salinity). Over regional scales, however, we only observed weak statistical relationships. All predictive algorithms we tested showed poor performance in predicting DMS concentrations across the NESAP region, although performance was improved through the use of regionally-tuned coefficients. Our compiled data set further support the importance of the NESAP as a global DMS 'hot spot' in summer, with patterns of DMS concentrations and sea–air fluxes similar to those observed in Lana et al.'s 2011 climatology. Given the significance of the NESAP in global oceanic DMS emissions, future studies should seek to improve mechanistic

understanding of the factors driving DMS accumulation in this region, with the aim of predicting climate-dependent changes
over the coming decades.

## Code availability

The codes used for spatial binning and data analysis can be provided by the authors upon request.

## Data availability

All previously publicly unavailable DMS concentration data presented here have been submitted to the NOAA PMEL database
(http://saga.pmel.noaa.gov/dms/). Ancillary shipboard and satellite data can be provided by the authors upon request.

## Author contributions

A. Herr compiled and analysed all data presented here, wrote all MATLAB codes, and wrote the manuscript, with editing and
intellectual input provided by P. Tortell R. Kiene and J. Dacey. R. Kiene further provided all biological rate measurement
data presented here, and J. Dacey assisted in collection of other field measurements.

## Competing interests

The authors declare that they have no conflict of interest.

## Acknowledgements

We dedicate this article to the memory of Dr. Ron Kiene, a wonderful scientist, mentor and friend. His contributions to DMS/P
research have shaped our field over the past three decades, and he will be missed by many around the world. We also wish to
thank many individuals involved in data collection and logistical aspects of the cruises presented here, including scientists
from the Institute of Ocean Sciences, the captain and crew of the R/V *Oceanus* and the CCGS *John P. Tully*, and members
Tortell, Kiene, Levine and Hatton laboratory groups. We also thank T. Ahlvin for GIS support, and both reviewers for their
insightful comments. Support for this work was provided from the US National Science Foundation (Grant #1436344), and
from the Natural Sciences and Engineering Research Council of Canada.

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

16  **Table 1. Summary of DMS data included in this study.  With the exception of the PMEL data, all measurements are derived from**

17  **membrane inlet mass spectrometry (MIMS).**

| Cruise abbreviation | Vessel affiliation; cruise name and number | Sampling dates | Areal extent | Provinces included | No. data points | References |
|---|---|---|---|---|---|---|
| VIJ04 | DFO; Central Coast BioChemical Study; 2004-24 | 12–19 Aug 2004 | 48˚ N - 52˚ N 131 ˚ W - 123˚ W | ALSK, CCAL | 1913 | (Nemcek et al. 2008) |
| LPJ07 | DFO; Line P; 2007-13 | 1–16 Jun 2007 | 47˚ N - 55˚ N 146 ˚ W - 123˚ W | ALSK, CCAL, PSAE | 21478 | (Asher et al. 2011) |
| LPA07 | DFO; Line P; 2007-15 | 16–30 Aug 2007 | 48˚ N - 54˚ N 146 ˚ W - 123˚ W | ALSK, CCAL, PSAE | 16418 | (Asher et al. 2011) |
| LPJ08 | DFO; Line P; 2008-26 | 1–15 Jun 2008 | 48˚N - 52˚N 146 ˚ W - 123˚ W | ALSK, CCAL, PSAE | 15304 | (Asher et al. 2011) |
| LPA08 | DFO; Line P; 2008-27 | 14–30 Aug 2008 | 48˚N - 52˚N 146 ˚ W - 123˚ W | ALSK, CCAL, PSAE | 20881 | (Asher et al. 2011) |
| VIJ10 | DFO; La Perouse; 2010-12 | 1–4 Jun 2010 | 48˚N - 52˚N 130 ˚ W - 123˚ W | ALSK, CCAL | 4551 | (Tortell et al. 2012) |

| | | | | | | |
|---|---|---|---|---|---|---|
| WCAC10 | DFO; Ocean Acidification; 2010-36 | 22 Jul–15 Aug 2010 | 47˚N - 57˚N 138 ˚ W - 123 ˚ W | ALSK, CCAL, PSAE | 25167 | (Asher et al. 2017) |
| LPA11 | DFO; Line P; 2011-27 | 19–28 Aug 2011 | 48˚ N - 51˚ N 146 ˚ W - 126˚ W | CCAL, PSAE | 10802 | (Asher et al. 2017) |
| LPA14 | DFO; Line P and Strait of Georgia; 2014-19 | 29–31 Aug 2014 | 50˚ N - 51˚ N 145˚ W - 134˚ W | PSAE | 2560 | (Asher et al. 2015) |
| O16 | UNOLS; Resolving DMS 1: OC-1607A | 12–27 Jul 2016 | 45˚ N - 56˚ N 143 ˚ W - 124˚ W | ALSK, CCAL, PSAE | 18712 | Previously unpublished |
| O17 | UNOLS; Resolving DMS II: OC-1708A | 12–27 Aug 2017 | 47˚ N - 57˚ N 146 ˚ W - 126˚ W | ALSK, CCAL, PSAE | 10015 | Previously unpublished |
| PMEL | various | various, 1984–2004 | 45˚ N - 61˚ N 167 ˚ W - 124˚ W | ALSK, CCAL, PSAE | 3236 | Various |

**Table 2. Summertime DMS data coverage across the NESAP region and within Longhurst provinces. Values indicate the number**
**of data-containing 1° x 1° spatial bins out of the total number of bins within the given area, with percent coverage of area shown in**
**parentheses. The left column represents the coverage for the PMEL data set (as utilized by L11) and the right column represents**
**the updated data set containing both PMEL measurements and MIM-based DMS concentration measurements.**

| Province Name | PMEL | This Study |
|---|---|---|
| CCAL | 30/75 (40.0 %) | 45/75 (60.0 %) |
| ALSK | 61/119 (51.3 %) | 83/119 (69.8 %) |
| PSAE | 5/430 (12.8 %) | 114/430 (26.5 %) |
| Total | 126/1140 (11.1 %) | 249/1140 (21.8 %) |

1  **Table 3. Mean DMS concentrations, sea-air fluxes and total summertime DMS flux for the PMEL data set utilized by L11, and the**
2  **updated data based used in this study.**

| | PMEL | | | This Study | | |
|---|---|---|---|---|---|---|
| **Province Name** | DMS (nM) | DMS Flux ($\mu$mol m$^{-2}$ d$^{-1}$) | Total summer DMS flux (Tg S) | DMS (nM) | DMS Flux ($\mu$mol m$^{-2}$ d$^{-1}$) | Total summer DMS flux (Tg S) |
| **CCAL** | $4.0 \pm 0.5$ | $4.4 \pm 0.95$ | 0.01 | $4.6 \pm 0.4$ | $6.3 \pm 0.7$ | 0.02 |
| **ALSK** | $8.9 \pm 1.1$ | $16.4 \pm 4.0$ | 0.06 | $7.5 \pm 0.9$ | $14.4 \pm 3.0$ | 0.05 |
| **PSAE** | $8.9 \pm 0.7$ | $21.0 \pm 4.0$ | 0.38 | $6.5 \pm 0.4$ | $16.5 \pm 2.2$ | 0.30 |
| **Total** | $7.2 \pm 0.5$ | $12.7 \pm 2.0$ | 0.32 | $6.2 \pm 0.3$ | $12.2 \pm 1.4$ | 0.30 |

1 **Table 4. Pearson's correlation coefficients between DMS concentrations and other oceanographic variables binned to 1° spatial**
2 **resolution. DMS data were derived from our combined PMEL and MIMS data set, variables derived from in-situ and satellite-**
3 **based data. N represents the number of data pairs available for each correlation calculation. * indicates significance of p<0.05.**

| Variable | Whole region | CCAL | ALSK | PSAE |
|---|---|---|---|---|
| Salinity | r = -0.04 | r = 0.24 | r = -0.04 | r = 0.07 |
|  | N = 223 | N = 31 | N = 83 | N = 102 |
| SST | r = -0.01 | r = -0.17 | r = -0.32* | r = 0.18 |
|  | N =248 | N = 44 | N = 83 | N = 114 |
| Chlorophyll-*a* | r = 0.17* | r = -0.11 | r = 0.34* | r = 0.45* |
|  | N =207 | N = 31 | N = 79 | N = 99 |
| Calcite | r = 0.12 | r = -0.08 | r = -0.01 | r = 0.50* |
|  | N =205 | N = 30 | N = 83 | N = 99 |
| PAR | r = 0.04 | r = -0.28 | r = 0.41* | r = 0.19 |
|  | N = 212 | N = 32 | N =52 | N = 91 |
| Depth | r = -0.05 | r = 0.20 | r = -0.34* | r = -0.02 |
|  | N = 201 | N = 45 | N = 12 | N = 96 |
| MLD | r = -0.14 | r = 0.14 | r = -0.06 | r = -0.18 |
|  | N = 98 | N = 21 | N = 11 | N = 70 |
| SSN | r = 0.01 | r = 0.14 | r = 0.30* | r = -0.18 |
|  | N = 207 | N = 31 | N = 79 | N = 99 |
| SSHA | r = -0.20* | r = -0.34 | r = -0.05 | r = -0.47* |
|  | N = 207 | N = 30 | N = 80 | N = 102 |
| NCP | r = 0.22* | r = 0.43* | r = 0.05 | r = 0.29 |
|  | N = 91 | N = 26 | N = 25 | N = 37 |
| Wind | r = 0.17* | r = -0.06 | r = 0.08 | r = 0.29* |
|  | N = 249 | N = 45 | N = 83 | N = 114 |

**Table 5. Pearson correlation coefficients and root mean square errors (nmol L⁻¹) between observed DMS concentrations and empirical predictions derived**
**from the SD02, VS07 and W07 algorithms, using both published coefficients (original) and coefficients derived specifically for our NESAP observations**
**using a least-squares approach (custom). Algorithm performance is shown for full NESAP region, as well as the three Longhurst biogeographical**
**provinces within our study area. * indicates significance of p<0.05.**

| Province | SD02 original | SD02 custom | VS07 original | VS07 custom | W07 original | W07 custom | G18 original | G18 custom |
|---|---|---|---|---|---|---|---|---|
| Whole region | r = 0.05 | r = 0.08 | r = -0.31* | r = 0.31* | r = -0.08 | r = 0.17* | r = 0.19 | r = 0.26* |
| | RMSE = 3.77 | RMSE = 3.03 | RMSE = 4.95 | RMSE = 2.63 | RMSE = 67.1 | RMSE = 5.86 | RMSE = 3.1 | RMSE = 19.8 |
| CCAL | r = 0.04 | r = 0.62* | r = -0.23 | r = 0.23 | r = -0.17 | r = 0.27 | r = 0.72* | r = 0.69* |
| | RMSE = 3.42 | RMSE = 1.61 | RMSE = 4.54 | RMSE = 1.20 | RMSE = 81.6 | RMSE = 2.04 | RMSE = 1.9 | RMSE = 55.2 |
| ALSK | r = 0.16 | r = 0.12 | r = -0.10 | r = 0.10 | r = -0.20 | r = 0.53* | r = -0.41 | r = 0.56 |
| | RMSE = 2.37 | RMSE = 2.07 | RMSE = 3.43 | RMSE = 2.09 | RMSE = 47.5 | RMSE = 7.19 | RMSE = 3.4 | RMSE = 31.17 |
| PSAE | r = 0.09 | r = 0.23 | r = -0.39* | r = 0.39* | r = -0.01 | r = 0.44* | r = -0.04 | r = 0.26 |
| | RMSE = 3.97 | RMSE = 2.94 | RMSE = 5.28 | RMSE = 2.81 | RMSE = 20.6 | RMSE = 4.59 | RMSE = 3.5 | RMSE = 21.3 |

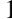

**Figure 1. Cruise tracks and discrete sampling stations (red circles) for the July 2016 (O16) cruise (a,c) and August 2017 (O17) cruise (b, d). Panels (a) and (b) show chl-*a* concentration (log scale), derived from AquaMODIS satellite, and averaged over the duration of the respective cruise. Panels (c) and (d) show average sea surface height anomaly (SSHA). Panel (a) shows the location of the T1-T3 transects surveyed during the 2016, whereas panel (b) shows the geographic location of locations of interest. The grey line represents the coastal-oceanic boundary, defined here as the 2000 m isobath.**

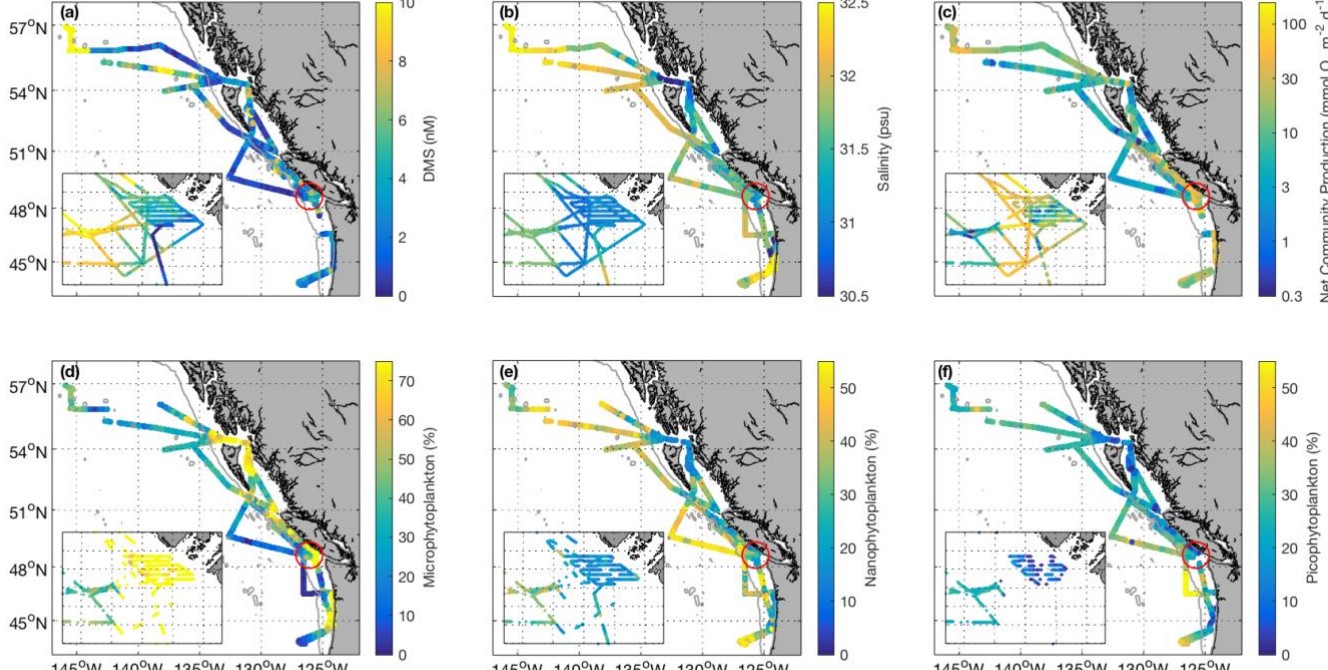

Figure 2. Spatial distribution of DMS (a), salinity (b), net community production (C; note log scale), and micro-, nano-, and picophytoplankton relative abundance (d-f) during the O16 cruise July of 2016 and the O17 cruise August of 2017. Colour scaling on the maps are adjusted to ensure readability and best illustrate spatial patterns. Some data values are higher than the maximum scale of the colour bar. The inset box shows the La Perouse Bank region, as marked by the red circle. The grey line represents the coastal-open ocean boundary (2000 m isobath).

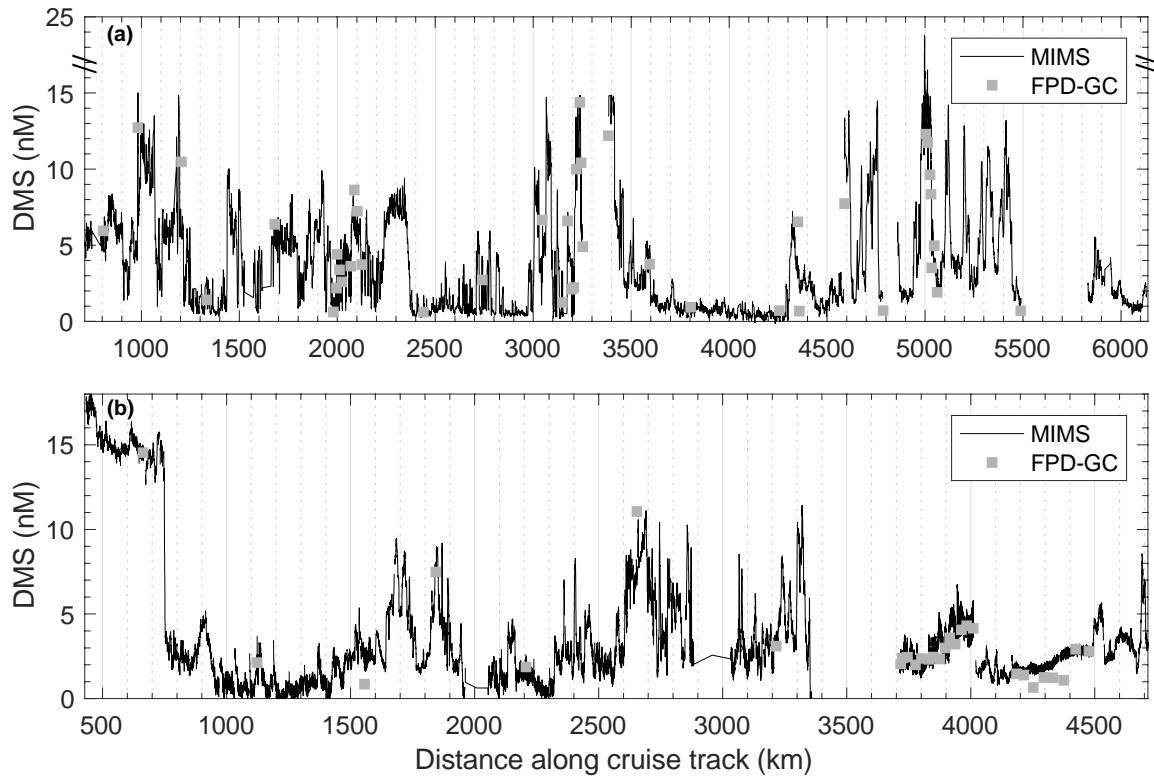

Figure 3. DMS concentrations during the O16 cruise in July of 2016 (a) and the O17 cruise in August of 2017 (b) as measured by membrane inlet mass spectrometry (MIMS, continuous black line) and a purge-and-trap sampling system connected to a gas chromatograph equipped with a flame-photometric detector (grey symbols). Mean absolute error was 0.93 nM and root mean squared error was 1.4 nM for all paired measurements between the two instruments. A linear regression of the two data sets yields a coefficients of determination of $r^2=0.89$.

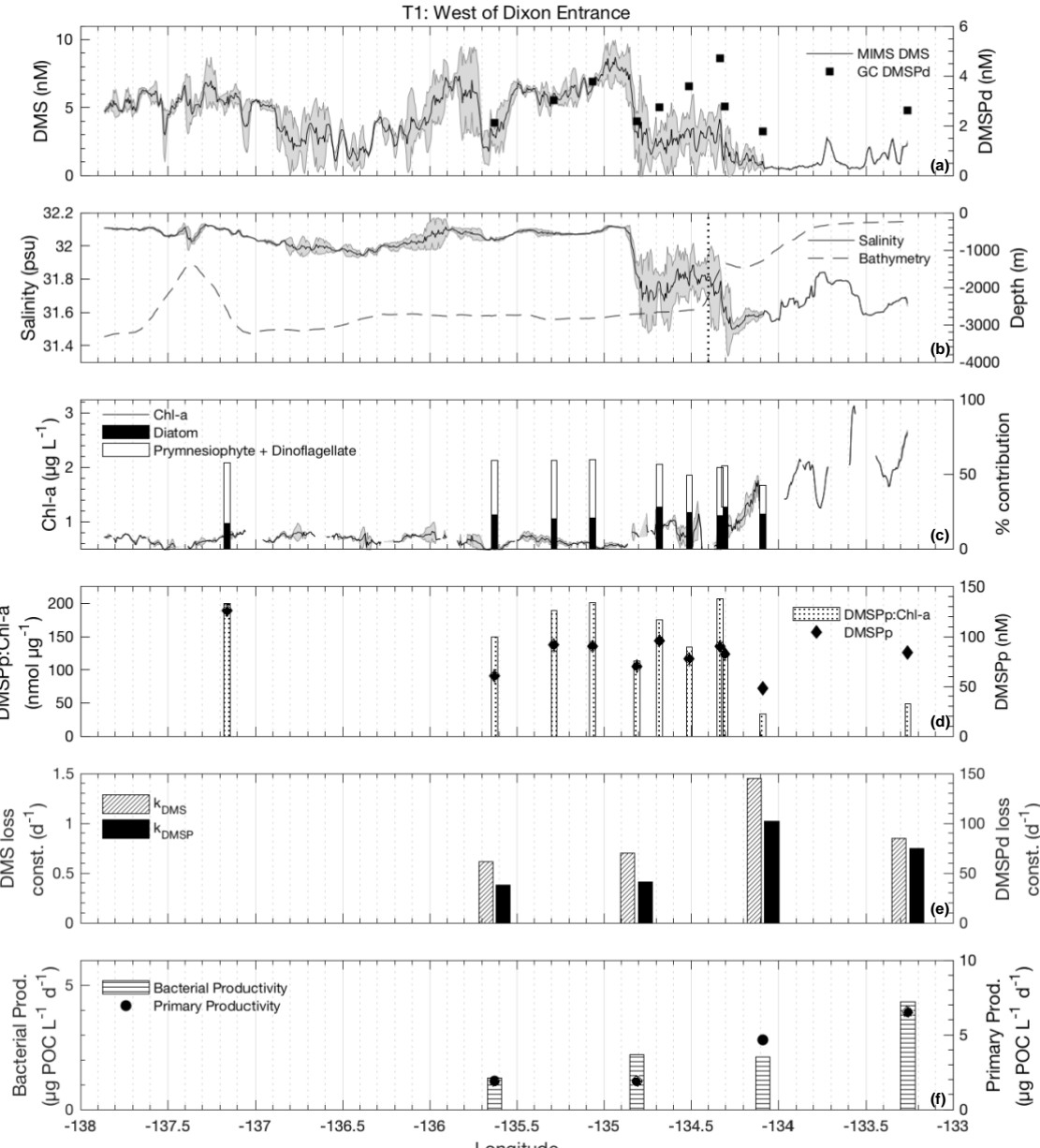

**Figure 4. MIMS-based DMS concentration measurements and station-based DMSPd measurements (a), salinity and bathymetry**
**(b), chl-*a* and HPLC-based station estimates of diatom and prymnesiophyte as defined % contribution to total assemblage (c),**
**DMSPp concentrations and DMSPp:chl-*a* ratios (d), DMS/P consumption rate constants (e), and bacterial and primary productivity**
**rates (f) along the T1 transect west of Dixon Entrance during July 2016 (O16 cruise).  Shaded regions represent standard deviation**
**of repeated measurements across the transect.  The vertical dotted line in panel (b) indicates the approximate shelf break (2000 m),**
**at 134.4˚ W.**

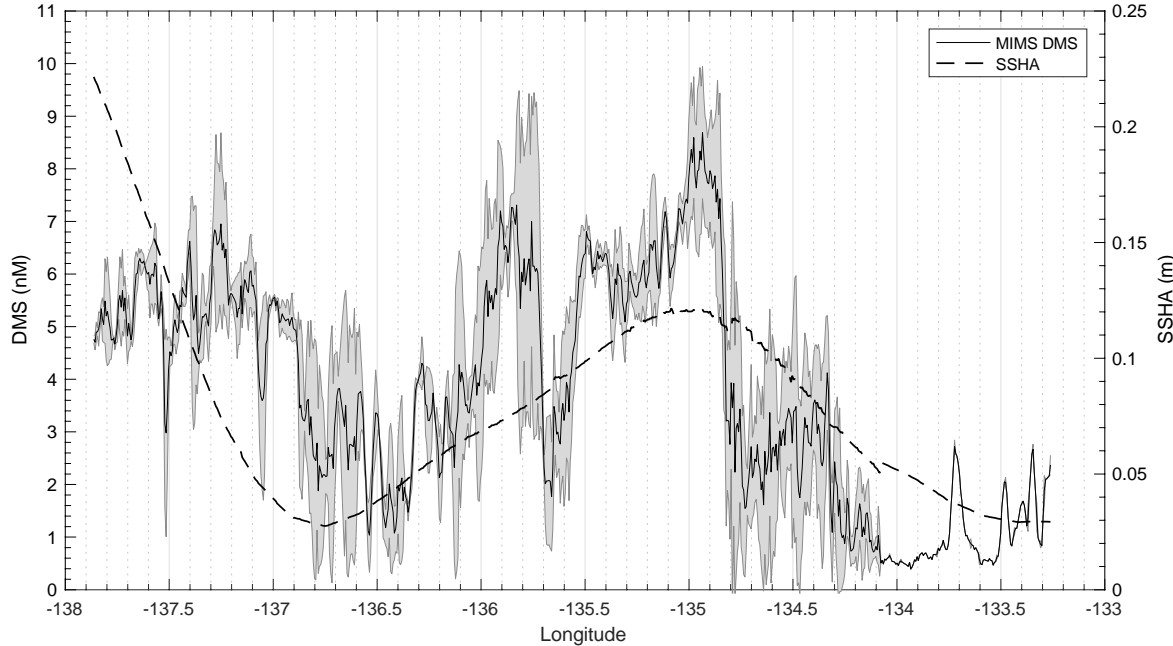

**Figure 5.** **Line plot of sea surface height anomaly (SSHA) on 15 July, 2016 and observed DMS concentrations between 14 July and 16 July, 2016 along T1. DMS along the T1 transect is highest in those areas influenced by positive SSHA values.**

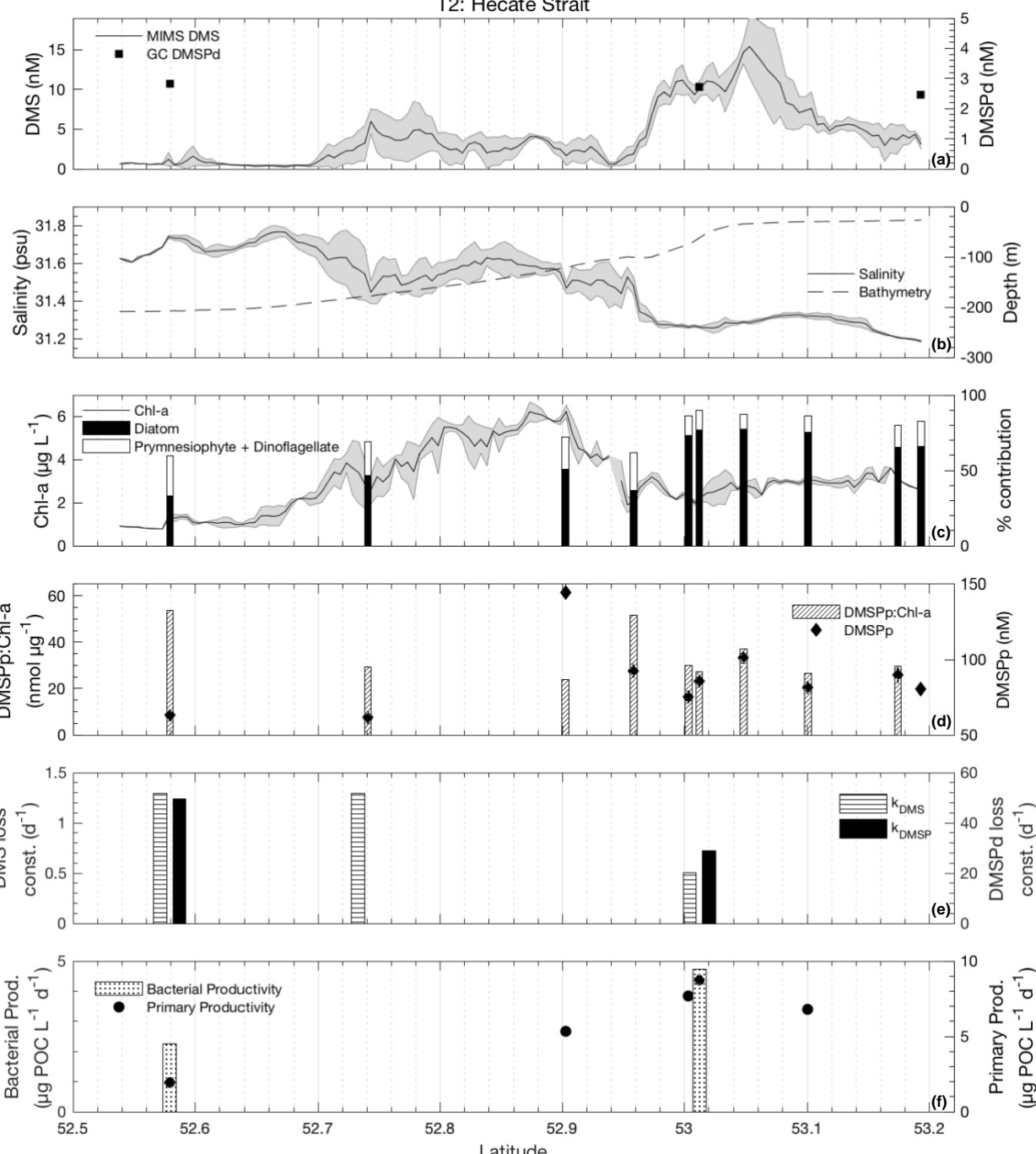

2    **Figure 6. As for Fig. 4, but for the T2 transect.**

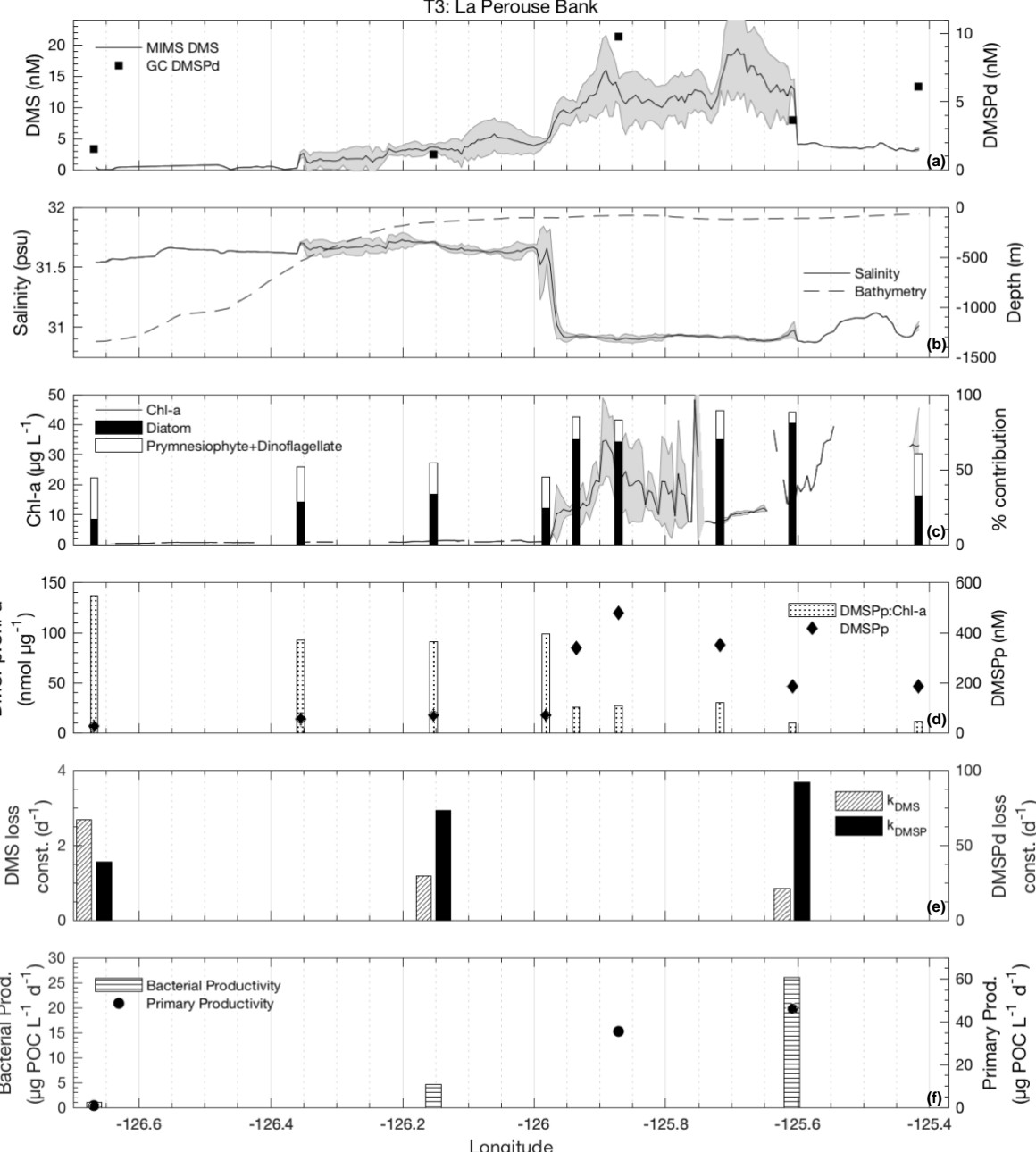

4  **Figure 7. As for Figs. 4 and 6, but for the T3 transect.**

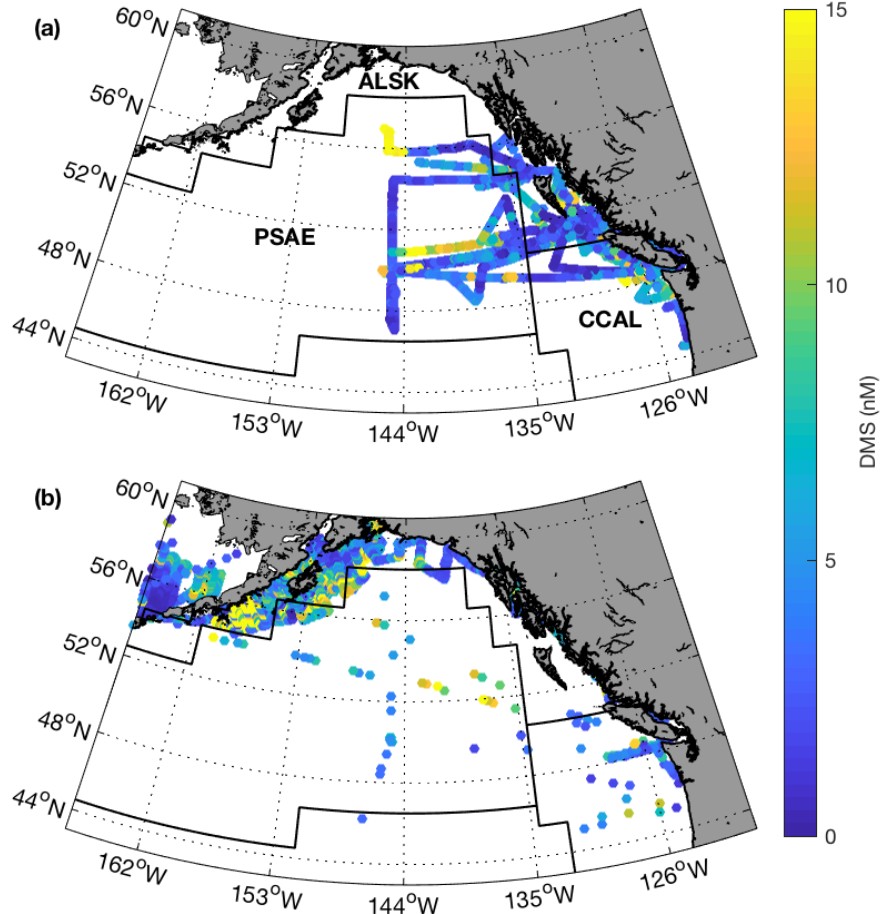

**Figure 8. Spatial distribution of summertime DMS measurements from MIMS (a; 2004-2017) and the PMEL (b; 1984 - 2004) data set. Black lines represent boundaries of Longhurst biogeographical provinces, with province names show in panel (a).**

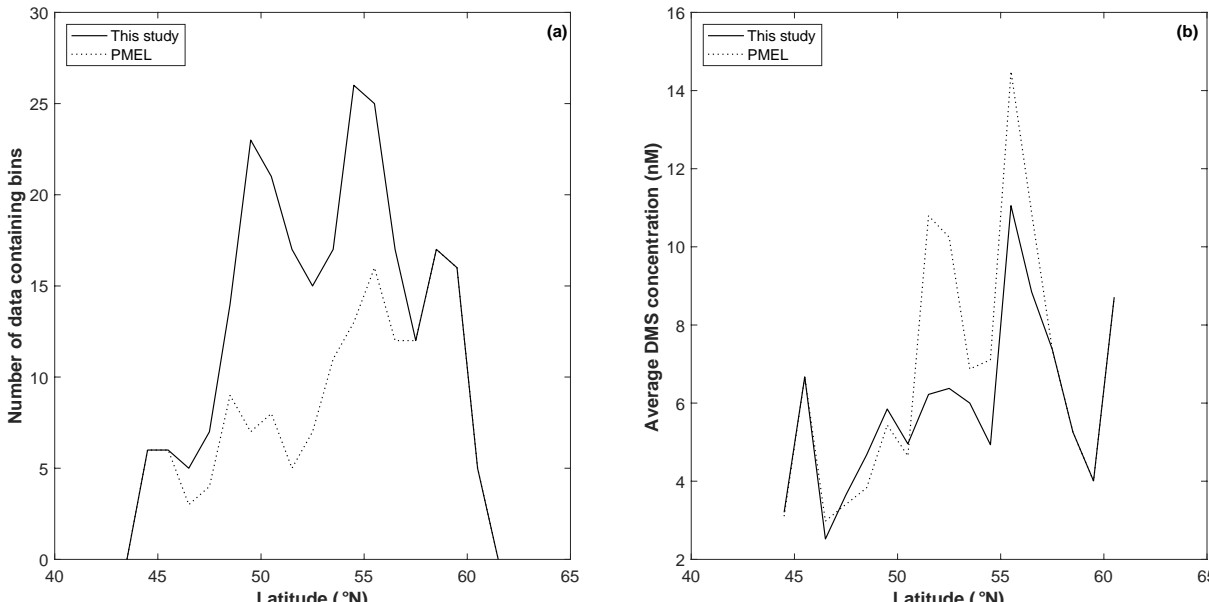

2 **Figure 9. Latitudinal distribution of data-containing bins (a) and average DMS concentration (b) for PMEL (dotted line) and**
3 **combined (PMEL and MIMS) data sets.**

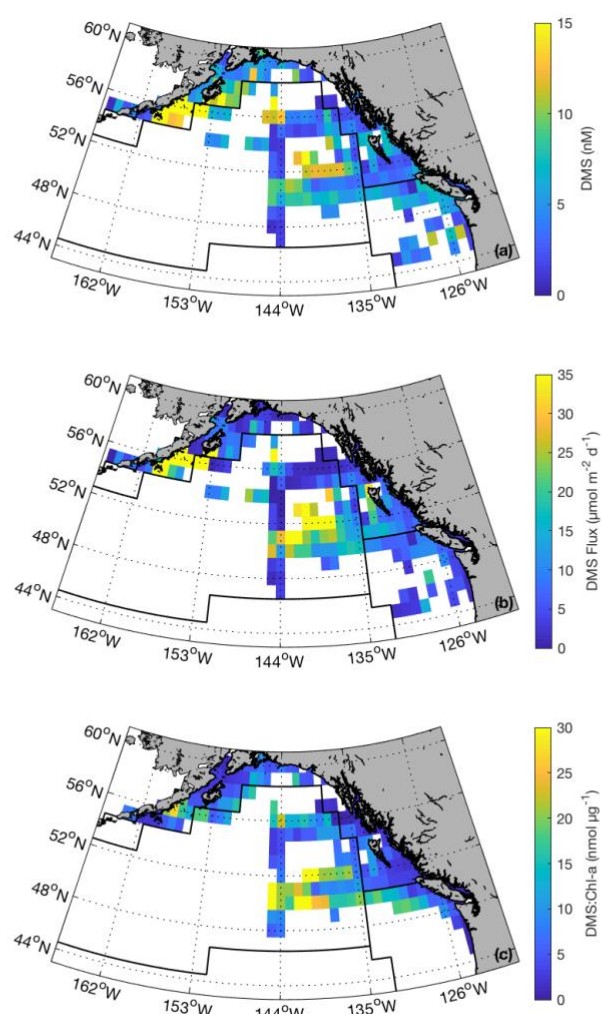

**Figure 10. Summertime DMS concentrations (a), DMS sea-air fluxes (b), and DMS:chl-*a* ratios (c) binned to 1° x 1° spatial resolution. These maps were derived using our combined PMEL/MIMS data set (1984–2017; June, July and August). Black lines correspond to boundaries of Longhurst biogeochemical provinces (see Fig. 8 for province names). Maximum values (47 nM, 180 μmol m⁻² d⁻¹, and 47 nmol μg⁻¹ for panels a, b, and c, respectively) exceed the bounds of the colorbars. Maximum values for DMS and DMS flux occur in the waters south of the Alaska Peninsula, whereas maximum DMS:chl-*a* occurs mid PSAE.**

