# Peer review of "Patterns and drivers of dimethylsulfide concentration in the northeast Subarctic Pacific across multiple spatial and temporal scales"

_Biogeosciences, 2018_

## Referee Comment (RC1) · M. Galí (Referee) · 6 Nov 2018

**Review: "Patterns and drivers of dimethylsulfide concentration in the northeast Subarctic Pacific across multiple spatial and temporal scales"**

Alysia E. Herr, Ronald P. Kiene, John W. H. Dacey, Philippe D. Tortell

Reviewer: Martí Galí

**General comments**

DMS accounts for approximately 20% of global sulfur emissions and represents a major source of cloud-seeding aerosols in unpolluted marine atmospheres. Therefore, marine DMS emission plays a key climatic role by modulating the radiative properties of aerosols and clouds, as well as precipitation. However, our understanding of DMS drivers across multiple spatial and temporal scales remains limited, and thus our predictive capacity. Reliable high-resolution DMS datasets are essential to improve regional and global DMS climatologies, avoiding artifacts and biases that affect interpolated climatologies based on sparse data, and potentially providing sufficient observations to allow for the evaluation of interannual variability.

The paper by Alysia E. Herr and coauthors makes a valuable contribution to the understanding of DMS distribution patterns in the northeast Subarctic Pacific (NESAP) region. Particularly, it highlights the difficulty in finding unifying criteria across this region, characterized by sharp biogeochemical gradients. From a methodological standpoint, I appreciated the authors presenting traditional discrete GC-FPD measurements along with high-resolution MIMS data (Fig. 3). This comparison increases our confidence in high-resolution DMS surveys, which might be prone to measurement artifacts caused by cell breakage upon pumping and in-line filtration. (I also appreciated the new datasets being readily uploaded to the PMEL archive!). The paper is well written and structured, and the figures and tables are clear and informative. Yet, I suggest the authors to add several citations to give readers a broader perspective on the subject, while recognizing the relevance of previous studies. Below I list what are, in my opinion, the main shortcomings of the article, which I recommend addressing through the Results, Discussion and Conclusions sections:

1. Treatment of DMSPt and phytoplankton groups/size classes in the data analysis. "Dominance vs. abundance":

1.1. I strongly encourage the authors to show total DMSP (DMSPt) concentrations, not just DMSPt:Chl ratios, and analyze their relationship with DMS, simply because DMSPt is the precursor of DMS. DMSPt concentration should be displayed (Fig. 4, 6 and 7).

1.2. Correlations between [DMS] and the dominance (relative abundance) of certain phytoplankton groups, as reported in this paper, can be misleading (the same applies to DMS vs. the DMSPt:Chl ratios). For example, in transect T3 (Fig. 7), the authors report a negative correlation between DMS and relative prymnesiophyte abundance (r = -0.75). In the middle of T3, inshore of the front, Chl increases sharply from ~1 to 30 μg L$^{-1}$, whereas % prymnesiophytes decreases from ~20% to ~5%. Still, the abundance of prymnesiophytes increases by about 8-fold, while DMS increases "only" by ~3-fold (~5

to ~15 nM). Thus, the increase in prymnesiophytes might suffice to explain the increase in DMS, be it through direct DMS release by prymnesiophytes or through the activity of micrograzers and bacteria. It is well known that increases in microphytoplankton abundance (mostly diatoms) are generally accompanied by increases of other phytoplankton groups (Barber and Hiscock, 2006; Uitz et al., 2006), as in the current dataset.

1.3 Dinoflagellates: Why is their abundance not reported and their role not discussed, although they are quoted in the Introduction and beginning of the Discussion as important players (Steiner et al., 2012)? More generally, I wonder what phytoplankton groups made up the ~50% of the pigment biomass that is omitted in Fig. 4, 6 and 7. What about other high-DMSP nanophytoplankton like chrysophytes and pelagophytes?

2. Process measurements:

2.1. I was puzzled to see dissolved DMSP consumption rate constants (kDMSPd) ranging between ~30 and 100 $d^{-1}$, while kDMSPd values in the literature are generally lower than 10 $d^{-1}$ (Galí and Simó, 2015). Highest kDMSPd reported so far are ~20 $d^{-1}$ in the NE Pacific (Royer et al., 2010) and near South Zealand (Lizotte et al., 2017). Is there a mistake? Everything would look more consistent in terms of S and C cycling (Kiene and Linn, 2000) if the reported kDMSPd had units of $h^{-1}$ instead of $d^{-1}$. Otherwise, the contribution of heterotrophic DMSPd conusmption to DMSPt cycling and bacterial S and C demand would be suspiciously high (as some quick calculations can show).

2.2. Although the data suggest distinct DMS(P) cycling regimes, I am not sure the amount of process measurements suffices to resolve DMS variability across fronts. In addition, important DMS production and loss terms were not measured. The finding that variations in biological DMS consumption (kDMS) drove [DMS] across regions seems robust (although DMS photolysis, a potentially important loss term, was not measured). Regarding DMS sources, DMS production from particulate DMSP cleavage was not assessed, and previous studies suggest it is important globally (Galí and Simó, 2015) and in this region (Asher et al., 2017). This fits well with prymnesiophyte and (dinoflagellate?) driven DMS production.
As DMS concentration is set by the dynamic balance between gross production rates (nM $d^{-1}$) and total DMS loss rate constants ($d^{-1}$) (Galí and Simó, 2015), conclusions based on a subset of production and loss processes are weak. Note also that the contribution of DMSPd turnover to DMS concentration is set by the product [DMSPd]*KDMSPd*Yd, where Yd is the DMS yield from dissolved DMSPd consumption. Thus, the relationship between kDMSPd and [DMS] tells little if Yd is not known (Yd can easily vary between 5 and 20%). The control of kDMS on [DMS] is comparatively more direct. I propose:

(i) displaying the relationship between kDMS, [DMS] and potentially other variables (T, S, Chl, ...) in a scatterplot-like graphic.

(ii) discussing more in depth the control of [DMS] by measured and non-measured DMS budget terms to give a more balanced view of the potential processes at play (see Galí and Simó, 2015).

3. Algorithm evaluation and regional tuning:

Several previous studies have shown that global-scale algorithms have poor skill at predicting regional DMS variability (e.g. Bell et al., 2006; Galí et al., 2018; Hind et al., 2011), not to talk about the mesoscale. The authors tried to tune pre-existing algorithms to their dataset using least squares fits, but they did not show the new coefficients, only the improved skill metrics. Consequently, the interpretation of this exercise remains vague and does not shed much light on the controlling factors, nor it helps designing better algorithms. I suggest either deleting this section (my choice) or reshaping and expanding it to make it more informative (making use of the supplemental information).

**Specific comments**

Introduction

P2 L17: please specify what zone of the "Southern Ocean". Iron-depleted regions of the subpolar Southern Ocean (approx. 40 to 60S) are relatively unproductive and typically have low [DMS]... (Jarníková and Tortell, 2016; Kiene et al., 2007; Lana et al., 2011).

P3 L1-7: These lines suggest that we do understand what causes high DMS concentrations in this area. I think we don't, for two main reasons:
(i) We do not understand interannual variability (Galí et al., 2018; Steiner et al., 2012).
(ii) We do not understand well enough the interplay between iron limitation, dominance of high DMSP producers (depicted by DMSPt:Chl) and DMS production pathways. The authors quote "the effects of mixed layer stratification and Fe-limitation, which may act to increase DMS/P production as a means to offset oxidative stress (Sunda et al. 2002)". However, Royer et al. (2010) found a positive correlation between iron concentration and DMSPt:Chl ratios in the HNLC area within the NESAP. This is contrary to what one would expect if Fe stress caused major increases in DMSPt:Chl ratios. It is possible that small Fe additions stimulate preferentially high DMSP producers (Levasseur et al., 2006; Steiner et al., 2012), which would result in a positive correlation between DMSPt:Chl and Fe as long as high DMSP producers are not outcompeted by diatoms. Regarding DMSP-to-DMS yields, although Royer et al. (2010) documented higher bacterial DMS yields in the Fe-depleted HNLC region, Steiner et al. (2012) pointed to dinoflagellates and micrograzers as key players in DMS production. The latter would imply a dominant role of DMS production from the particulate pool, involving phytoplanktonic DMSP lyases. These processes are poorly documented in the NESAP area.

P3 L12: I suggest citing here Simó et al. (2018) (The quantitative role of microzooplankton grazing in dimethylsulfide (DMS) production in the NW Mediterranean) to support the importance of grazing-mediated DMS production. A major

finding of that paper is that "throughout the year, grazing-mediated DMS production explained 73% of the variance in the DMS concentration".

P3 L18. I missed two relevant citations here:
1. Belviso et al. (2003) "Mesoscale features of surface water DMSP and DMS concentrations in the Atlantic Ocean off Morocco and in the Mediterranean Sea". A precursor study showing sharp changes in DMS:DMSPt:Chl across mesoscale features and fronts.
2. Royer et al. (2015) Small-scale variability patterns of DMS and phytoplankton in surface waters of the "tropical and subtropical Atlantic, Indian, and Pacific Oceans". A high-resolution DMS survey across 21000 km in the tropical oceans, showing that "much of the variability in DMS concentrations occurs at scales between 15 and 50 km, that is, at the lower edge of mesoscale dynamics, decreasing with latitude and productivity. DMS variability was found to be more commonly related to that of phytoplankton-related variables than to that of physical variables".

Methods

P5 L3: Please report more statistics (RMSE, mean bias, linear regression equation...) comparing MIMS and GC-FPD, either here or on the figure (which should be 3, not 2).

P6 L27: Phytoplankton biomass tends to peak in late summer in the oceanic sector of the NESAP. Regarding DMS, there seems to be more similarity between August and September than between August and June (Galí et al., 2018; Lana et al., 2011; Steiner et al., 2012), perhaps related to the dinoflagellate abundance in late summer (Steiner et al., 2012). I understand the authors' choice reflects data availability, but I suggest cautioning the reader that June, July and August can be very different.

P7 L14-15: Please add citations for all satellite products: PAR (Frouin et al., 2003), Chl-a (Hu et al., 2012; O'Reilly et al., 1998).

P7 L29: What fraction of your 1-degree bins is over bottom depths shallower than 2000 m, where MLD is not available? Could this bias the evaluation of empirical algorithms, given the distinct biogeochemical dynamics of shelf seas? What would be the impact of replacing time-resolved MLD data by a monthly climatology? Perhaps using climatological MLD would make a little difference in oceanic areas, while allowing testing of the algorithms in the entire NESAP domain. Also, how did data gaps caused by cloudiness affect algorithm evaluation? (as suggested by P9 L25 in Results, and Fig. 1 top panels).

P7 L30: Please specify what DMS and SST datasets were used in combination with daily 2.5-degree wind speed data to calculate DMS flux: non-binned data, 1-degree binned data retaining temporal variability, or the 1-degree summer climatology?

P8 L9: I would add "all observations within the JJA months for a given year were averaged" (to avoid confusion with the way monthly climatologies like L11 are calculated).

P8 L27: Did you try Spearman's rank correlations? This could help identifying nonlinear monotonic relationships.

P8 L32: If the section on algorithm evaluation is not dropped, the authors could also evaluate the two-step algorithm of Galí et al. (2018). We showed that it outperforms SD02 and VS07 in most oceanic areas, although it has difficulties to reproduce the DMS seasonal cycle at Ocean Station P. The global algorithm of Anderson et al. (2001) would also be an interesting choice here as it performed well across contrasting trophic regimes in the SE Pacific (Hind et al., 2011).

Results

P10 L10-15: How well compare the estimates of phytoplankton size classes derived from absorption (underway WetLabs instrument) to HPLC pigments?

P10 L18: The difference may be significant due to large N, but is it relevant when the ranges overlap so much? Note also that saying DMS concentrations were significantly different in different years downplays the utility of calculating a multiyear climatology.

P11 L12: Beyond the high SSHA-DMS coherence at the mesoscale, Fig. 5 shows there is a lot of unresolved submesoscale variability, probably due to biological heterogeneity, in agreement with (Royer et al., 2015).

P11 L20: Relative abundance or absolute? Also, I suggest specifying that these size classes were derived from underway absorption data, not HPLC, to avoid confusion.

P11 L30: How can significance be tested with n = 2 at each side of the front (Fig. 4)? I suggest reporting ranges. Qualitatively, I agree that k's were different at either side of the front.

P12 L1-2: This explanation is unclear. See general comment 2.2.

P12 L31-32: Same as above.

P13 L30: Can this be tested statistically somehow? See general comment 2.2.

P13 L33: Please check units (general comment 2.1).

P15 L15-26: See general comment 3. This section would be more interesting if the authors explained the rationale behind the original algorithms, reported the tuned coefficients, and explained in what sense they alter the performance of the original

algorithms. In particular, turning from negative to positive slope in VS07 completely alters the rationale behind this algorithm.

Discussion

P17 L13: could you please explain more explicitly the relationship between positive SSHA and high DMS? Eg, anticyclonic eddies detaching from a frontal area and transporting a certain water type, or whatever... This can help us understand why SSHA can show both positive and negative correlations to DMS (as explained at the bottom of the same page).

P17 L19: The unpublished meta-analysis should be cited as pers. comm., I guess.

P17 L23-27: The negative correlation between SRD and DMS in the whole region (Table 5, original VS07 algorithm) goes against this argument.

P18 L20: OK, but transient [DMS] and kDMS do not even need to be invoked. The relationship may also arise at nearly-steady state, where [DMS] = (gross production rate) / (total loss rate constant), where total loss is dominated by biological DMS consumption k, as explained by (Galí and Simó, 2015).

P18 L26: As the authors explain, DMS consumption rates will be positively correlated to [DMS] as long as [DMS] is more variable than kDMS, because cons. rate = [DMS]*kDMS. I suggest removing this sentence.: "In contrast to DMS rate constants (d-1), water column DMS consumption rates (nM d-1) showed a positive correlation with DMS concentrations (r=0.65, p=0.01). This result is not unexpected, as consumption rates are the product of rate constants and in situ concentrations".

P19 L4: I also suggest removing this: "In contrast to biological loss, turnover time due to sea-air flux showed no correlation to DMS concentrations".

P19 L13-23: I suggest refining the writing here: see general comment 2.2.

P19 section 4.4: Would it be possible to quantify interannual variability of DMS (see Fig. 3 of Steiner et al., 2012) using the merged dataset? Interannual variability has been overlooked (Galí et al., 2018), with so much emphasis on the mean (climatological) state... This part of the Discussion is currently a bit poor.

P20-21, section 4.6: The last two paragraphs of this section seem to contradict each other. Does elevated productivity (which usually follows elevated biomass) translate into high DMS, or not? If only in some places, why? What are the relevant scales for this comparison? It would be interesting to cite here the work of (Kameyama et al., 2013) "Strong relationship between dimethyl sulfide and net community production in the western subarctic Pacific", perhaps extracting more information from your own NCP vs. DMS data.

P21 L17-18: Data from tables 4 and 5 supports this idea, so I suggest citing the tables here (ie, the HNCL PSAE shows the highest negative correlations between DMS and bot NO3 and SRD).

**Edits**

P5 L5-6: I suggest removing "and rate measurements to examine potential drivers of spatial variation". Rates were not measured in underway samples...

P6 L12: "sampled" should be "samples".

Table 1: please replace "June" by "August" for cruise LPA07.

P12 L7-8: please remove "given no new production". DMS removal expressed as a daily % would also hold in the presence of DMS production (as it is usually the case).

P15 L1: Please correct "We also calculated and DMS:Chla..."

P20 L14: "distinction", rather than "measure"?

**Reviewer references**

Anderson, T. R., Spall, S. A., Yool, A., Cipollini, P., Challenor, P. G. and Fasham, M. J. R.: Global fields of sea surface dimethylsulfide predicted from chlorophyll, nutrients and light, J. Mar. Syst., 30, 1–20, 2001.

Asher, E., Dacey, J. W., Ianson, D., Peña, M. A. and Tortell, P. D.: Concentrations and cycling of DMS, DMSP, and DMSO in coastal and offshore waters of the Subarctic Pacific during summer, 2010-2011, J. Geophys. Res. Ocean., 119, 7123–7138, doi:10.1002/2014JC010066, 2017.

Barber, R. T. and Hiscock, M. R.: A rising tide lifts all phytoplankton: Growth response of other phytoplankton taxa in diatom-dominated blooms, Global Biogeochem. Cycles, 20(4), 1–12, doi:10.1029/2006GB002726, 2006.

Bell, T., Malin, G., Mckee, C. and Liss, P.: A comparison of dimethylsulphide (DMS) data from the Atlantic Meridional Transect (AMT) programme with proposed algorithms for global surface DMS concentrations, Deep Sea Res. Part II Top. Stud. Oceanogr., 53(14–16), 1720–1735, doi:10.1016/j.dsr2.2006.05.013, 2006.

Belviso, S., Sciandra, A. and Copin-Montégut, C.: Mesoscale features of surface water DMSP and DMS concentrations in the Atlantic Ocean off Morocco and in the Mediterranean Sea, Deep. Res. Part I Oceanogr. Res. Pap., 50, 543–555, doi:10.1016/S0967-0637(03)00032-3, 2003.

Frouin, R., Franz, B. and Wang, M.: Algorithm to estimate PAR from SeaWiFS data Version 1.2-Documentation., 2003.

Galí, M. and Simó, R.: A meta-analysis of oceanic DMS and DMSP cycling processes: Disentangling the summer paradox, Global Biogeochem. Cycles, 29, 496–515, doi:10.1002/2014GB004940, 2015.

Galí, M., Levasseur, M., Devred, E., Simó, R. and Babin, M.: Sea-surface dimethylsulfide (DMS) concentration from satellite data at global and regional scales, Biogeosciences, 15, 3497–3519, doi:10.5194/bg-15-3497-2018, 2018.

Hind, A. J., Rauschenberg, C. D., Johnson, J. E., Yang, M. and Matrai, P. A.: The use of algorithms to predict surface seawater dimethyl sulphide concentrations in the SE Pacific, a region of steep gradients in primary productivity , biomass and mixed layer depth, Biogeosciences, doi:10.5194/bg-8-1-2011, 2011.

Hu, C., Lee, Z. and Franz, B.: Chlorophyll *a* algorithms for oligotrophic oceans: A novel approach based on three-band reflectance difference, J. Geophys. Res., 117(C1), C01011, doi:10.1029/2011JC007395, 2012.

Jarníková, T. and Tortell, P. D.: Towards a revised climatology of summertime dimethylsulfide concentrations and sea-air fluxes in the Southern Ocean, Environ. Chem., 13(2), 364–378, doi:10.1071/EN14272, 2016.

Kameyama, S., Tanimoto, H., Inomata, S., Yoshikawa-Inoue, H., Tsunogai, U., Tsuda, A., Uematsu, M., Ishii, M., Sasano, D., Suzuki, K. and Nosaka, Y.: Strong relationship between dimethyl sulfide and net community production in the western subarctic Pacific, Geophys. Res. Lett., 40(15), 3986–3990, doi:10.1002/grl.50654, 2013.

Kiene, R. P. and Linn, L. J.: Distribution and Turnover of Dissolved DMSP and Its Relationship with Bacterial Production and Dimethylsulfide in the Gulf of Mexico, Limnol. Oceanogr., 45(4), 849–861, 2000.

Kiene, R. P., Kieber, D. J., Slezak, D., Toole, D. A., Valle, D. del, Bisgrove, J., Brinkley, J. and Rellinger, A.: Distribution and cycling of dimethylsulfide, dimethylsulfoniopropionate, and dimethylsulfoxide during spring and early summer in the Southern Ocean south of New Zealand, Aquat. Sci., 69(3), 305–319, doi:10.1007/s00027-007-0892-3, 2007.

Lana, A., Bell, T. G., Simó, R., Vallina, S. M., Ballabrera-Poy, J., Kettle, A. J., Dachs, J., Bopp, L., Saltzman, E. S., Stefels, J., Johnson, J. E. and Liss, P. S.: An updated climatology of surface dimethlysulfide concentrations and emission fluxes in the global ocean, Global Biogeochem. Cycles, 25, GB1004, doi:10.1029/2010GB003850, 2011.

Levasseur, M., Scarratt, M., Michaud, S., Merzouk, A., Wong, C., Arychuk, M., Richardson, W., Rivkin, R., Hale, M. and Wong, S.: DMSP and DMS dynamics during a

mesoscale iron fertilization experiment in the Northeast Pacific—Part I: Temporal and vertical distributions, Deep Sea Res. Part II Top. Stud. Oceanogr., 53(20–22), 2353–2369, doi:10.1016/j.dsr2.2006.05.023, 2006.

Lizotte, M., Levasseur, M., Law, C. S., Walker, C. F., Safi, K. A., Marriner, A. and Kiene, R. P.: Dimethylsulfoniopropionate (DMSP) and dimethyl sulfide (DMS) cycling across contrasting biological hotspots of the New Zealand subtropical front, Ocean Sci., 13(6), 961–982, doi:10.5194/os-13-961-2017, 2017.

O'Reilly, J. E., Maritorena, S., Mitchell, B. G., Siegel, D. A., Carder, K. L., Garver, S. A., Kahru, M. and McClain, C.: Ocean color chlorophyll algorithms for SeaWiFS, J. Geophys. Res., 103(C11), 24937–24953, doi:10.1029/98JC02160, 1998.

Royer, S., Mahajan, A. S., Galí, M., Saltzman, E. and Simó, R.: Small-scale variability patterns of DMS and phytoplankton in surface waters of the tropical and subtropical Atlantic, Indian, and Pacific Oceans, Geophys. Res. Lett., 42, 475–483, doi:10.1002/2014GL062543, 2015.

Royer, S.-J., Levasseur, M., Lizotte, M., Arychuk, M., Scarratt, M. G., Wong, C., Lovejoy, C., Robert, M., Johnson, K., Pena, A., Michaud, S. and Kiene, R. P.: Microbial dimethylsulfoniopropionate (DMSP) dynamics along a natural iron gradient in the northeast subarctic Pacific, Limnol. Oceanogr., 55(4), 1614–1626, doi:10.4319/lo.2010.55.4.1614, 2010.

Simó, R., Saló, V., Almeda, R., Movilla, J., Trepat, I., Saiz, E. and Calbet, A.: The quantitative role of microzooplankton grazing in dimethylsulfide (DMS) production in the NW Mediterranean, Biogeochemistry, 2, 1–18, doi:10.1007/s10533-018-0506-2, 2018.

Steiner, N. S., Robert, M., Arychuk, M., Levasseur, M. L., Merzouk, A., Peña, M. A., Richardson, W. A. and Tortell, P. D.: Evaluating DMS measurements and model results in the Northeast subarctic Pacific from 1996–2010, Biogeochemistry, 110(1–3), 269–285, doi:10.1007/s10533-011-9669-9, 2012.

Uitz, J., Claustre, H., Morel, A. and Hooker, S. B.: Vertical distribution of phytoplankton communities in open ocean: An assessment based on surface chlorophyll, J. Geophys. Res., 111(C8), doi:10.1029/2005JC003207, 2006.

---

## Referee Comment (RC2) · Anonymous Referee #2 · 25 Nov 2018

As the title indicates, this manuscript describes DMS and DMSP spatial and temporal distribution patterns with high-resolution field data collected in coastal waters of the NE Pacific. The DMS patterns are compared to historical field data in a publicly available database, collected mostly in previous decades and further offshore. Measurements of some of the rates involved in DMS cycling allow for interpreatation and discussion of the likely controls (or drivers) of the next patterns observed. No single physical or biological parameter accounted for the DMS/P variability observed and described as a whole in the region; rather probable controls change in relative importance within subregions, as has been shown in other studies. This variability only confirms the already described complexity of the DMS/P/O biogeochemical system at any one place and

time. . The manuscript is very well written and is a pleasure to read. . However, the authors should decide whether they will focus this report on the NE Pacific ONLY and hence solely references for this region will be used. Right now, the manuscript ignores many references to similar conclusions in other regions or even in nearby SE Bering Sea (Barnard et al.) while occasionally using references from other regions to support its own conclusions (eg. North Sea, Southern Ocean). The authors miss a unique chance to strengthen the conclusions of this manuscript. .  Summary of Comments on bg-2018-411 review.pdf Page: 3 -Line 1: what is L11? Lana et al. 2011? Please check throughout ms -Line 18: Holligan et al 1987 first reported the link between DMS and fronts; even if it was not in NESAP waters but NAtlantic waters . Page: 6 -line 7: Please report BP data in carbon units, not leucine units so they can be compared with PP data and with other studies. -Line 8: Hence, as done previously by Kiene et al, DMSPp can be estimated such that the DMSP/chl ratios are estimated with both parameters in the particulate fraction; only makes a difference where and when [DMSPd] are high. Fig 4 shows a match for DMSPt and DMSPd measurements; hence, DMSPp can be calculated. -line 9: with a GC-FPD discrete method -line 12: sampleS -line 15: "The estimation formulas" used? . Page: 7 -line 6: where were the SSS and SST matches obtained from? The PMEL data set does not provide them. . Page: 8 -line 21-22: I had come to assume that L11= Lana et al 2011. If yes, please reword this sentence -line 23: please insert "The PMEL" data were first... -line 32: replace 'that' with 'those' . Page: 10 -similar DMS/P-NPP relationship by Bell et al for the North and South Atlantic along the AMT transect and by Matrai et al for the Barents Sea. Should be addressed in the Discussion. . Page: 11 -2nd paragraph: because similar conclusions of prymensiophytes vs other phyto groups and DMS/P patterns were drawn by Barnard et al 1984? in the SE Bering Sea, they should be definitively mentioned in the Discussion. . Page: 15 -line 1: something is missing before 'and'; or remove 'and'; or replace by 'a"? . Page: 16 -section 4.1 and elsewhere: Since references beyond the NESAP are already included, other -mostly older- very pertinent references have been suggested in this review and should be included to strengthen

the arguments made. -line 11: but not in polar waters (Turner et al for southern ocean; Matrai et al for Barents Sea) -line 13: please insert after 'physiological state' ", as previously shown by Gabric et al. (1999)" [Barents Sea] -line 14: please insert 'e.g.' in front of the refs listed, as there are other pertinent refs as well -line 27: please insert 'and elsewhere' after NESAP. -line 31: which studies? add references! . Page: 17 -line 6: waterS -somewhere in this page: A similar conclusion on the influence of prymensiophytes in 3 coastal domains just a bit north in the NE Pacific was reported by Barnard et al 1984. Please include. -line 19: update the McParland and Levine ref, as the ms has moved on in its review process -line 26, after the Sunda et al. 2002 ref. Please address the observation that a post-bloom = also when bacterial activity is highest and DMSPd > DMSPp, as phyto cells become leakier (eg, Matrai and Keller 1993 and Malin et al 1993 for cocco blooms; Stefels et al review as well) . Page: 18 -line 3: add a few references after 'cell lysis' for all processes mentioned -line 6: instead and/or in addition to the variables reported herein? -line 8: which 'studies'? add refs (e.g., xxx) -line 9: is this only for coastal waters of the NESAP? Or elsewhere also? Please specify. This is not a new observation for other regions (e.g., Turner et al Southern Ocean, Matrai et al. Barents Sea) -line 21: please insert "in other regions" after 'previous studies'! -line 29: please insert "in other regions" after 'previous studies' -line 31-32: check punctuation . Page: 19 -line 2: please convert to carbon units! -line 7: it IS possible -line 21-23: delete this paragraph. It is naive and does not add anything . Page: 20 -section 4.5: Both Hind et al 2011 and Deutsch et al 2009 in the Eastern SPacific and globally, respectively, combined Longhurst provinces and DMS-based algorithms to test their predictions. Both studies should be referenced and included here as they discussed the strngths and weaknesses of such DMS predictive algorithms. Hind et al. also include many of the variables discussed in this study, even the presence of eddies and upwelling. . Page: 22 -line 1: supportS -line 1: please insert "in summer" after 'hotspots' -line 10: By US NSF rules, shouldn't all data be submitted to a long-term data repository? . Page: 24-32 References -check subscripts and italics for scientific names throughout -format references; remove all

caps throughout . Page: 34 -please insert "(in parenthesis)" at the end of the Table 2 title. That's what is in (xxx), right? . Page: 41 -Fig 4d y-axis: why not DMSPp/chl a? both are particle-bound variables These are discrete stations. -Fig 4f: can you please report BP in Carbon units? otherwise it cannot be compared with PP or other studies . Page: 42 -Fig 5: the y-axis scale is missing . Page: 43 -same comments as for Fig 4 . Page: 44 -same comments as for Fig 4 . Page: 46 -Fig 9: Given the tables, fig 8 and the fact that the differences in the DMS flux estimates is so small, this figure does not add much and could be removed

---

## Author Comment (AC1) · 30 Jan 2019

Response to Reviewer 1:

*Thank you for your thoughtful and comprehensive critique of our work. We appreciate your insight and attention to detail, and believe the edits made in response to your comments have strengthened our manuscript. We have followed many of your suggestions, including the inclusion of dinoflagellate abundance data, discussion (and statistics) on relative vs. total abundance of different phytoplankton groups, and discussion of the limitations of providing only a subset of process measurements. We also tested additional algorithms (including that of Galí et al. 2018) with some success. Below we provide a detailed response to all of the reviewer's comments (in italics), indicating the changes that have been made. Line numbers refer to those of the revised manuscript (supplement to this response), which includes all tracked changes.*

**General comments**

DMS accounts for approximately 20% of global sulfur emissions and represents a major source of cloud-seeding aerosols in unpolluted marine atmospheres. Therefore, marine DMS emission plays a key climatic role by modulating the radiative properties of aerosols and clouds, as well as precipitation. However, our understanding of DMS drivers across multiple spatial and temporal scales remains limited, and thus our predictive capacity. Reliable high-resolution DMS datasets are essential to improve regional and global DMS climatologies, avoiding artifacts and biases that affect interpolated climatologies based on sparse data, and potentially providing sufficient observations to allow for the evaluation of interannual variability.

The paper by Alysia E. Herr and coauthors makes a valuable contribution to the understanding of DMS distribution patterns in the northeast Subarctic Pacific (NESAP) region. Particularly, it highlights the difficulty in finding unifying criteria across this region, characterized by sharp biogeochemical gradients. From a methodological standpoint, I appreciated the authors presenting traditional discrete GC-FPD measurements along with high-resolution MIMS data (Fig. 3). This comparison increases our confidence in high-resolution DMS surveys, which might be prone to measurement artifacts caused by cell breakage upon pumping and in-line filtration. (I also appreciated the new datasets being readily uploaded to the PMEL archive!). The paper is well written and structured, and the figures and tables are clear and informative. Yet, I suggest the authors to add several citations to give readers a broader perspective on the subject, while recognizing the relevance of previous studies. Below I list what are, in my opinion, the main shortcomings of the article, which I recommend addressing through the Results, Discussion and Conclusions sections:

1. Treatment of DMSPt and phytoplankton groups/size classes in the data analysis. "Dominance vs. abundance":

1.1. I strongly encourage the authors to show total DMSP (DMSPt) concentrations, not just DMSPt:Chl ratios, and analyze their relationship with DMS, simply because DMSPt is the precursor of DMS. DMSPt concentration should be displayed (Fig. 4, 6 and 7).

*We have added DMSPp data to Figs. 4, 6, and 7, and have converted DMSPt:Chl to DMSPp:Chl (as per the second referee's request). DMSPd is already included in Figs. 4, 6 and 7, and thus*

*DMSPt can be deduced. We have also examined statistical relationships between DMS, DMSPp and DMSPp:Chl: pg 12, line 2, 26-28; pg 13 line 20-22; pg 14, line 2; pg 45; pg 47-48.*

1.2. Correlations between [DMS] and the dominance (relative abundance) of certain phytoplankton groups, as reported in this paper, can be misleading (the same applies to DMS vs. the DMSPt:Chl ratios). For example, in transect T3 (Fig. 7), the authors report a negative correlation between DMS and relative prymnesiophyte abundance (r = -0.75). In the middle of T3, inshore of the front, Chl increases sharply from ~1 to 30 µg L$^{-1}$, whereas % prymnesiophytes decreases from ~20% to ~5%. Still, the abundance of prymnesiophytes increases by about 8-fold, while DMS increases "only" by ~3-fold (~5 to ~15 nM). Thus, the increase in prymnesiophytes might suffice to explain the increase in DMS, be it through direct DMS release by prymnesiophytes or through the activity of micrograzers and bacteria. It is well known that increases in microphytoplankton abundance (mostly diatoms) are generally accompanied by increases of other phytoplankton groups (Barber and Hiscock, 2006; Uitz et al., 2006), as in the current dataset.

*Thank you for pointing this out. We calculated total abundance of various phytoplankton groups and found no correlations between DMS and total prymnesiophyte and dinoflagellate abundance. Along T3, correlation between DMS and total diatom abundance remained quite high. We have included discussion and some statistics regarding this point: pg 18, line 1-7.*

1.3 Dinoflagellates: Why is their abundance not reported and their role not discussed, although they are quoted in the Introduction and beginning of the Discussion as important players (Steiner et al., 2012)? More generally, I wonder what phytoplankton groups made up the ~50% of the pigment biomass that is omitted in Fig. 4, 6 and 7. What about other high-DMSP nanophytoplankton like chrysophytes and pelagophytes?

*We initially did not include dinoflagellate data, as this group represented only a minor proportion (<10%) of phytoplankton across our study area. In the revised article, we now discuss the relationship between DMS and the combined relative abundance of prymnesiophytes and dinoflagellates. In general, results from this modified analysis are very similar to those obtained from a prymnesiophyte-only approach, with increased correlation coefficients in some cases: pg 11, line 33; pg 13, line 3; pg 14, line 1.*

*We have also included more data regarding other phytoplankton groups, including green algae, picoeukaryotes and prokaryotes: pg 11, line 26; pg 12, line 29-30; pg 13, line 30-31.*

*As our HPLC-derived estimates were somewhat limited with regards to taxa specificity, we unfortunately do not have information on the abundance of chrysophytes or pelagophytes.*

2. Process measurements:

2.1. I was puzzled to see dissolved DMSP consumption rate constants (kDMSPd) ranging between ~30 and 100 d$^{-1}$, while kDMSPd values in the literature are generally lower than 10 d$^{-1}$ (Galí and Simó, 2015). Highest kDMSPd reported so far are ~20 d$^{-1}$ in the NE Pacific (Royer et al., 2010) and near South Zealand (Lizotte et al., 2017). Is there a mistake? Everything would

look more consistent in terms of S and C cycling (Kiene and Linn, 2000) if the reported kDMSPd had units of $h^{-1}$ instead of $d^{-1}$. Otherwise, the contribution of heterotrophic DMSPd conusmption to DMSPt cycling and bacterial S and C demand would be suspiciously high (as some quick calculations can show).

*The rate constants for DMSPd turnover were indeed high in the study area, and they are, in fact, among the highest measured anywhere.  These measurements were made by Ron Kiene's group, one of the world-leaders in this area.  Ron carefully examined the raw data and was confident that the measurements are robust.  (The reviewer may know that Ron tragically passed away very recently).  The very high rate constants were observed were likely due, in part, to the high productivity waters sampled, as well as some methodological modifications that minimized release of DMSPd during the $^{35}$S-DMSP tracer incubations.  We now briefly mention these factors in our revised discussion: pg 5, line 30; pg 20, line 4-5.*

2.2. Although the data suggest distinct DMS(P) cycling regimes, I am not sure the amount of process measurements suffices to resolve DMS variability across fronts. In addition, important DMS production and loss terms were not measured. The finding that variations in biological DMS consumption (kDMS) drove [DMS] across regions seems robust (although DMS photolysis, a potentially important loss term, was not measured). Regarding DMS sources, DMS production from particulate DMSP cleavage was not assessed, and previous studies suggest it is important globally (Galí and Simó, 2015) and in this region (Asher et al., 2017). This fits well with prymnesiophyte and (dinoflagellate?) driven DMS production.

As DMS concentration is set by the dynamic balance between gross production rates (nM $d^{-1}$) and total DMS loss rate constants ($d^{-1}$) (Galí and Simó, 2015), conclusions based on a subset of production and loss processes are weak. Note also that the contribution of DMSPd turnover to DMS concentration is set by the product [DMSPd]*KDMSPd*Yd, where Yd is the DMS yield from dissolved DMSPd consumption. Thus, the relationship between kDMSPd and [DMS] tells little if Yd is not known (Yd can easily vary between 5 and 20%). The control of kDMS on [DMS] is comparatively more direct. I propose:

(i) displaying the relationship between kDMS, [DMS] and potentially other variables (T, S, Chl, ...) in a scatterplot-like graphic.

(ii) discussing more in depth the control of [DMS] by measured and non-measured DMS budget terms to give a more balanced view of the potential processes at play (see Galí and Simó, 2015).

*This is a fair point.  To address it, we now discuss the limitations of the conclusions we can draw from our data, acknowledging the lack of measurement of several important terms: pg 19, lines 4-10.*

*We made some scatter plots, as suggested by the reviewer, but these plots only showed the moderate linear relationship between kDMS and [DMS], with no particular trend based on temperature, salinity, etc., thus adding little information.*

3. Algorithm evaluation and regional tuning:

Several previous studies have shown that global-scale algorithms have poor skill at predicting regional DMS variability (e.g. Bell et al., 2006; Galí et al., 2018; Hind et al., 2011), not to talk about the mesoscale. The authors tried to tune pre-existing algorithms to their dataset using least squares fits, but they did not show the new coefficients, only the improved skill metrics. Consequently, the interpretation of this exercise remains vague and does not shed much light on the controlling factors, nor it helps designing better algorithms. I suggest either deleting this section (my choice) or reshaping and expanding it to make it more informative (making use of the supplemental information).

*We have clarified our discussion of algorithm results, and also included results from the new Galí et al., 2018 algorithm (see responses below): pg 15, lines 29-34; pg 22, lines 7-14.*

*Coefficients for each algorithm were recalculated for each region tested, resulting in 16 sets of coefficients. Since these regionally-tuned algorithms performed poorly, we see little utility in reporting them.*

**Specific comments**

Introduction

P2 L17: please specify what zone of the "Southern Ocean". Iron-depleted regions of the subpolar Southern Ocean (approx. 40 to 60S) are relatively unproductive and typically have low [DMS]... (Jarníková and Tortell, 2016; Kiene et al., 2007; Lana et al., 2011).

*Done: pg 2, line 16-17*

P3 L1-7: These lines suggest that we do understand what causes high DMS concentrations in this area. I think we don't, for two main reasons:
(i) We do not understand interannual variability (Galí et al., 2018; Steiner et al., 2012). (ii) We do not understand well enough the interplay between iron limitation, dominance of high DMSP producers (depicted by DMSPt:Chl) and DMS production pathways. The authors quote "the effects of mixed layer stratification and Fe-limitation, which may act to increase DMS/P production as a means to offset oxidative stress (Sunda et al. 2002)". However, Royer et al. (2010) found a positive correlation between iron concentration and DMSPt:Chl ratios in the HNLC area within the NESAP. This is contrary to what one would expect if Fe stress caused major increases in DMSPt:Chl ratios. It is possible that small Fe additions stimulate preferentially high DMSP producers (Levasseur et al., 2006; Steiner et al., 2012), which would result in a positive correlation between DMSPt:Chl and Fe as long as high DMSP producers are not outcompeted by diatoms. Regarding DMSP- to-DMS yields, although Royer et al. (2010) documented higher bacterial DMS yields in the Fe-depleted HNLC region, Steiner et al. (2012) pointed to dinoflagellates and micrograzers as key players in DMS production. The latter would imply a dominant role of DMS production from the particulate pool, involving phytoplanktonic DMSP lyases. These processes are poorly documented in the NESAP area.

*We have added some additional text clarifying the role of Fe limitation in driving DMS/P dynamics in the Subarctic Pacific: pg 3, lines 13-17.*

P3 L12: I suggest citing here Simó et al. (2018) (The quantitative role of microzooplankton grazing in dimethylsulfide (DMS) production in the NW Mediterranean) to support the importance of grazing-mediated DMS production. A major finding of that paper is that "throughout the year, grazing-mediated DMS production explained 73% of the variance in the DMS concentration".

*Thank you for this suggestion. We have now cited this paper: pg 3, line 12..*

P3 L18. I missed two relevant citations here:
1. Belviso et al. (2003) "Mesoscale features of surface water DMSP and DMS concentrations in the Atlantic Ocean off Morocco and in the Mediterranean Sea". A precursor study showing sharp changes in DMS:DMSPt:Chl across mesoscale features and fronts.
2. Royer et al. (2015) Small-scale variability patterns of DMS and phytoplankton in surface waters of the "tropical and subtropical Atlantic, Indian, and Pacific Oceans". A high-resolution DMS survey across 21000 km in the tropical oceans, showing that "much of the variability in DMS concentrations occurs at scales between 15 and 50 km, that is, at the lower edge of mesoscale dynamics, decreasing with latitude and productivity. DMS variability was found to be more commonly related to that of phytoplankton-related variables than to that of physical variables".

*Thank you, added: pg 3, lines 21-22.*

Methods

P5 L3: Please report more statistics (RMSE, mean bias, linear regression equation...) comparing MIMS and GC-FPD, either here or on the figure (which should be 3, not 2).

*Done: pg 5, lines 4-5.*

P6 L27: Phytoplankton biomass tends to peak in late summer in the oceanic sector of the NESAP. Regarding DMS, there seems to be more similarity between August and September than between August and June (Galí et al., 2018; Lana et al., 2011; Steiner et al., 2012), perhaps related to the dinoflagellate abundance in late summer (Steiner et al., 2012). I understand the authors' choice reflects data availability, but I suggest cautioning the reader that June, July and August can be very different.

*We have clarified a sentence justifying our choice of JJA for the summer climatology, while acknowledging the point raised by the reviewer: pg 6, line 33; pg 7, lines 1-2..*

P7 L14-15: Please add citations for all satellite products: PAR (Frouin et al., 2003), Chl-a (Hu et al., 2012; O'Reilly et al., 1998).

*Done, thank you: pg 7, lines 19-20.*

P7 L29: What fraction of your 1-degree bins is over bottom depths shallower than 2000 m, where MLD is not available? Could this bias the evaluation of empirical algorithms, given the distinct biogeochemical dynamics of shelf seas? What would be the impact of replacing time-resolved MLD data by a monthly climatology? Perhaps using climatological MLD would make a little difference in oceanic areas, while allowing testing of the algorithms in the entire NESAP domain. Also, how did data gaps caused by cloudiness affect algorithm evaluation? (as suggested by P9 L25 in Results, and Fig. 1 top panels).

*We have added information regarding % of MLD coverage: pg 8, lines 1-2.*

*We did indeed assess the utility of using a number of different MLD climatologies.  (We actually assessed empirical algorithms and pairwise regressions using climatologies of all variables). However, in no case did a monthly MLD climatology yield a stronger relationship with DMS. We chose to consistently use time-matched data rather than mixing approaches.*

P7 L30: Please specify what DMS and SST datasets were used in combination with daily 2.5-degree wind speed data to calculate DMS flux: non-binned data, 1-degree binned data retaining temporal variability, or the 1-degree summer climatology?

*Reworded for clarity: pg 8, lines 7-9.*

P8 L9: I would add "all observations within the JJA months for a given year were averaged" (to avoid confusion with the way monthly climatologies like L11 are calculated).

*Done: pg 8, lines 16-17.*

P8 L27: Did you try Spearman's rank correlations? This could help identifying nonlinear monotonic relationships.

*Yes. However, results were only minimally different than Pearson's correlation coefficients, and in no case revealed substantially stronger relationships.*

P8 L32: If the section on algorithm evaluation is not dropped, the authors could also evaluate the two-step algorithm of Galí et al. (2018). We showed that it outperforms SD02 and VS07 in most oceanic areas, although it has difficulties to reproduce the DMS seasonal cycle at Ocean Station P. The global algorithm of Anderson et al. (2001) would also be an interesting choice here as it performed well across contrasting trophic regimes in the SE Pacific (Hind et al., 2011).

*Based on these suggestions, we evaluated and included results from Galí et al. (2018), and found that it was able to reproduce DMS with reasonably good accuracy in the CCAL province.  We expanded our discussion based on these results, and compared them to the negative correlations found using the VS07 algorithm: pg 15, lines 29-34; pg 22, lines 7-14.*

*We tested the algorithm of Anderson et al. (2001), and found that it  performed very poorly across all areas.*

Results

P10 L10-15: How well compare the estimates of phytoplankton size classes derived from absorption (underway WetLabs instrument) to HPLC pigments?

*We have now addressed this in the Methods section: pg 5, lines 13-14.*

P10 L18: The difference may be significant due to large N, but is it relevant when the ranges overlap so much? Note also that saying DMS concentrations were significantly different in different years downplays the utility of calculating a multiyear climatology.

*This is a good point. We have deleted this sentence.*

P11 L12: Beyond the high SSHA-DMS coherence at the mesoscale, Fig. 5 shows there is a lot of unresolved submesoscale variability, probably due to biological heterogeneity, in agreement with (Royer et al., 2015).

*We have addressed this in the discussion: pg 17, line 26-27.*

P11 L20: Relative abundance or absolute? Also, I suggest specifying that these size classes were derived from underway absorption data, not HPLC, to avoid confusion.

*Done: pg 11, line 30.*

P11 L30: How can significance be tested with n = 2 at each side of the front (Fig. 4)? I suggest reporting ranges. Qualitatively, I agree that k's were different at either side of the front.

*We have changed wording to report this as a qualitative result, with insufficient sampling to allow for statistical testing: pg 12, line 7-8.*

P12 L1-2: This explanation is unclear. See general comment 2.2.
P12 L31-32: Same as above.

*We have clarified the wording here: pg 12, lines 12-13 and addressed the subject in discussion: pg 19, lines 4-10.*

P13 L30: Can this be tested statistically somehow? See general comment 2.2.

*Too few data points for reliable statistics. We have changed wording to report this as a qualitative result: pg 14, line 6.*

P13 L33: Please check units (general comment 2.1).

*Units OK (see response to general comment 2.1).*

P15 L15-26: See general comment 3. This section would be more interesting if the authors explained the rationale behind the original algorithms, reported the tuned coefficients, and explained in what sense they alter the performance of the original algorithms. In particular, turning from negative to positive slope in VS07 completely alters the rationale behind this algorithm.

*This phenomenon and the rationale behind this algorithm is now addressed in the revised discussion: pg 22, lines 7-14.*

Discussion

P17 L13: could you please explain more explicitly the relationship between positive SSHA and high DMS? Eg, anticyclonic eddies detaching from a frontal area and transporting a certain water type, or whatever... This can help us understand why SSHA can show both positive and negative correlations to DMS (as explained at the bottom of the same page).

*Done: pg 17, lines 25-37.*

P17 L19: The unpublished meta-analysis should be cited as pers. comm., I guess.

*This paper is now published and the citation has been updated: pg 17, line 32.*

P17 L23-27: The negative correlation between SRD and DMS in the whole region (Table 5, original VS07 algorithm) goes against this argument.

*Addressed in discussion: pg 22, lines 7-14.*

P18 L20: OK, but transient [DMS] and kDMS do not even need to be invoked. The relationship may also arise at nearly-steady state, where [DMS] = (gross production rate) / (total loss rate constant), where total loss is dominated by biological DMS consumption k, as explained by (Galí and Simó, 2015).

*We feel that the original wording is clear, and in keeping with the manner that results are presented by the subsequently cited studies.*

P18 L26: As the authors explain, DMS consumption rates will be positively correlated to [DMS] as long as [DMS] is more variable than kDMS, because cons. rate = [DMS]*kDMS. I suggest removing this sentence.: "In contrast to DMS rate constants (d- 1), water column DMS consumption rates (nM d-1) showed a positive correlation with DMS concentrations (r=0.65, p=0.01). This result is not unexpected, as consumption rates are the product of rate constants and in situ concentrations".

*Done.*

P19 L4: I also suggest removing this: "In contrast to biological loss, turnover time due to sea-air flux showed no correlation to DMS concentrations".

*Done.*

P19 L13-23: I suggest refining the writing here: see general comment 2.2.

*We have added discussion addressing the limitations of basing conclusions on a subset of rate measurement data: pg 19, lines 4 – 10.*

P19 section 4.4: Would it be possible to quantify interannual variability of DMS (see Fig. 3 of Steiner et al., 2012) using the merged dataset? Interannual variability has been overlooked (Galí et al., 2018), with so much emphasis on the mean (climatological) state... This part of the Discussion is currently a bit poor.

*Only 18/216 grid cells contained at least 5 years of data, and many of these were not contiguous. Thus, little information could be gleaned from this approach.*

P20-21, section 4.6: The last two paragraphs of this section seem to contradict each other. Does elevated productivity (which usually follows elevated biomass) translate into high DMS, or not? If only in some places, why? What are the relevant scales for this comparison? It would be interesting to cite here the work of (Kameyama et al., 2013) "Strong relationship between dimethyl sulfide and net community production in the western subarctic Pacific", perhaps extracting more information from your own NCP vs. DMS data.

*This is addressed further in section 4.7: pg 22, lines 7-14.*

P21 L17-18: Data from tables 4 and 5 supports this idea, so I suggest citing the tables here (ie, the HNCL PSAE shows the highest negative correlations between DMS and bot NO3 and SRD).

*Added citation for table 5, but not table 4, as the relationship between DMS and $NO_3$ in the PSAE is not significant: pg 22, line 8.*

**Edits**

P5 L5-6: I suggest removing "and rate measurements to examine potential drivers of spatial variation". Rates were not measured in underway samples...

*Reworded for clarity: pg 5, line 7.*

P6 L12: "sampled" should be "samples".

*Done: pg 6, line 16.*

Table 1: please replace "June" by "August" for cruise LPA07.

*Done: pg 37*

P12 L7-8: please remove "given no new production". DMS removal expressed as a daily % would also hold in the presence of DMS production (as it is usually the case).

*Done.*

P15 L1: Please correct "We also calculated and DMS:Chla..." P20 L14: "distinction", rather than "measure"?

*Done: pg 21, line 3.*

---

## Author Comment (AC2) · 30 Jan 2019

*Response to Reviewer #2,*

*Thank you for your thoughtful critique of our work, and the suggestion of many helpful references. We have followed your suggestions in many places, including the conversion of bacterial productivity data to carbon units, inclusion of DMSPp data, and discussion of previous studies from areas outside of the NESAP. Below we provide a detailed response to all of the reviewer's comments, indicating the changes that have been made. Line numbers refer to those of the revised manuscript (supplement to this response), which includes all tracked changes.*
-Line 18: Holligan et al 1987 first reported the link between DMS and fronts; even if it was not in NESAP waters but NAtlantic waters .

*We added this reference and others: pg 3, line 21-22*

Page: 6 -line 7: Please report BP data in carbon units, not leucine units so they can be compared with PP data and with other studies.

*Done: pg 6, line 9-10; Fig. 4, 6, 7*

-Line 8: Hence, as done previously by Kiene et al, DMSPp can be estimated such that the DMSP/chl ratios are estimated with both parameters in the particulate fraction; only makes a difference where and when [DMSPd] are high. Fig 4 shows a match for DMSPt and DMSPd measurements; hence, DMSPp can be calculated.

*We have added DMSPp values to figures: pg 6, line 12-13; Fig. 4, 6, 7*

-line 9: with a GC-FPD discrete method

*Added: pg 6, line 10-11*

-line 12: sampleS

*Corrected: pg 6, line 16*

-line 15: "The estimation formulas" used?

*Corrected: pg 6, line 20*

 Page: 7 -line 6: where were the SSS and SST matches obtained from? The PMEL data set does not provide them.

*Nearly all PMEL data for this region provides matched SSS and SST.  The percentage of DMS data obtained from this source with matched SSS and SST values has been added: pg 7, lines 10-11.*

Page: 8 -line 21-22: I had come to assume that L11= Lana et al 2011. If yes, please reword this sentence

*Corrected: pg 8, line 31*

-line 23: please insert "The PMEL" data were first...

*Corrected: pg 8, line 31*

-line 32: replace 'that' with 'those'

*Corrected: pg 9, line 9*

Page: 10 -similar DMS/P-NPP relationship by Bell et al for the North and South Atlantic along the AMT transect and by Matrai et al for the Barents Sea. Should be addressed in the Discussion.

*This is a relatively minor result based on our high resolution underway data. We did not include this topic in the discussion, as we believe it would dilute our discussion of contrasting DMS cycling regimes.*

Page: 11 -2nd paragraph: because similar conclusions of prymensiophytes vs other phyto groups and DMS/P patterns were drawn by Barnard et al 1984? in the SE Bering Sea, they should be definitively mentioned in the Discussion.

*Thank you for pointing out this oversight. We have added a reference to this study on pg 17, line 5, and point out the similar results directly on pg 17, line 35; pg 18, line 1.*

Page: 15 -line 1: something is missing before 'and'; or remove 'and'; or replace by 'a"?

*Corrected: pg 15, line 11.*

Page: 16 -section 4.1 and elsewhere: Since references beyond the NESAP are already included, other -mostly older- very pertinent references have been suggested in this review and should be included to strengthen the arguments made.
-line 11: but not in polar waters (Turner et al for southern ocean; Matrai et al for Barents Sea)

*Have added references: pg 16, line 19.*

line 13: please insert after 'physiological state' ", as previously shown by Gabric et al. (1999)" [Barents Sea]

*Done: pg 16, line 22.*

-line 14: please insert 'e.g.' in front of the refs listed, as there are other pertinent refs as well

*Done: pg 16, line 23*

-line 27: please insert 'and elsewhere' after NESAP

*Done: pg 17, line 5*

-line 31: which studies? add references!

*Done: pg 17, line 5*

Page: 17 -line 6: waterS

*Corrected: pg 17, line 16*

-somewhere in this page: A similar conclusion on the influence of prymensiophytes in 3 coastal domains just a bit north in the NE Pacific was reported by Barnard et al 1984. Please include.

*Done: pg 17, line 35; pg 18, line 1..*

-line 19: update the McParland and Levine ref, as the ms has moved on in its review process –

*Corrected: pg 17, line 32*

-line 26, after the Sunda et al. 2002 ref. Please address the observation that a post-bloom = also when bacterial activity is highest and DMSPd > DMSPp, as phyto cells become leakier (eg, Matrai and Keller 1993 and Malin et al 1993 for cocco blooms; Stefels et al review as well)

*We have now added some brief discussion of this: pg 18, lines 17-19*

Page: 18 -line 3: add a few references after 'cell lysis' for all processes mentioned -line 6: instead and/or in addition to the variables reported herein? -line 8: which 'studies'? add refs (e.g., xxx)

*Done: pg 18, lines 12-14*

-line 9: is this only for coastal waters of the NESAP? Or elsewhere also? Please specify. This is not a new observation for other regions (e.g., Turner et al Southern Ocean, Matrai et al. Barents Sea)

*Done: pg 18, lines 31-32*

-line 21: please insert "in other regions" after 'previous studies'!

*Done: pg 19, line 18*

-line 29: please insert "in other regions" after 'previous studies'

*Done: pg 19, line 24*

-line 31-32: check punctuation

*Reworded: pg 19, 26-27*

Page: 19 -line 2: please convert to carbon units!

*Done: pg 19, line 29*

-line 7: it IS possible

*Corrected: pg 19, line 32*

-line 21-23: delete this paragraph. It is naive and does not add anything

*Done.*

Page: 20 -section 4.5: Both Hind et al 2011 and Deutsch et al 2009 in the Eastern SPacific and globally, respectively, combined Longhurst provinces and DMS-based algorithms to test their predictions. Both studies should be referenced and included here as they discussed the strngths and weaknesses of such DMS predictive algorithms. Hind et al. also include many of the variables discussed in this study, even the presence of eddies and upwelling.

*We have included the reference to Hind. Perhaps the reviewer is referring to the paper by Derevianko et al. 2009, on which Deutch was an author. This paper examines model performance, but does not include Longhurst provinces (it is a global study). We have added two other papers that do utilize provinces (Belviso et al. 2011, and Royer et al. 2015): pg 19, line 2.*

Page: 22 -line 1: supportS

*This section was otherwise edited based on the other referee's comments.*

-line 1: please insert "in summer" after 'hotspots'

*Done: pg 22, line 31*

-line 10: By US NSF rules, shouldn't all data be submitted to a long-term data repository? .

*We have the data prepared for submission to the PMEL data server. However, the site is currently unavailable due to the partial US government shut-down. These data included paired ancillary variables.*

Page: 24-32 References -check subscripts and italics for scientific names throughout -format references; remove all caps throughout

*Corrected.*

Page: 34 -please insert "(in parenthesis)" at the end of the Table 2 title. That's what is in (xxx), right?

*Corrected (should be clear what's in parentheses now): pg 38, lines 2-3.*

Page: 41 -Fig 4d y-axis: why not DMSPp/chl a? both are particle-bound variables These are discrete stations.

*We have now changed this to* DMSPp/chl a  *: pg 45, line 4*

-Fig 4f: can you please report BP in Carbon units? otherwise it cannot be compared with PP or other studies

*Corrected: pg 45*

Page: 42 -Fig 5: the y-axis scale is missing

*Corrected: pg 46*

Page: 43 -same comments as for Fig 4
Page: 44 -same comments as for Fig 4

*Corrected: pg 47, 48*

Page: 46 -Fig 9: Given the tables, fig 8 and the fact that the differences in the DMS flux estimates is so small, this figure does not add much and could be removed

*We have chosen to keep this figure, as it demonstrates areas where concentrations vary significantly from PMEL data.*